# *Candida utilis* yeast as a functional protein source for Atlantic salmon (*Salmo salar* L.): Local intestinal tissue and plasma proteome responses

Felipe Eduardo Reveco-Urzua[1⊙¤], Mette Hofossæter[2⊙], Mallikarjuna Rao Kovi [3], Liv Torunn Mydland[1], Ragnhild Ånestad[1], Randi Sørby[2], Charles McLean Press[2], Leidy Lagos[1], Margareth Øverland[1] *

**1** Department of Animal and Aquaculture Sciences, Faculty of Biosciences, Norwegian University of Life Sciences, Aas, Norway, **2** Department of Basic Sciences and Aquatic Medicine, Faculty of Veterinary Medicine, Norwegian University of Life Sciences, Oslo, Norway, **3** Department of Plant Sciences, Faculty of Biosciences, Norwegian University of Life Sciences, Aas, Norway

⊙ These authors contributed equally to this work.
¤ Current address: Cargill Aqua Nutrition North Sea, Bergen, Norway
* margareth.overland@nmbu.no

## Abstract

Microbial ingredients such as *Candida utilis* yeast are known to be functional protein sources with immunomodulating effects whereas soybean meal causes soybean meal-induced enteritis in the distal intestine of Atlantic salmon (*Salmo salar* L.). Inflammatory or immuno-modulatory stimuli at the local level in the intestine may alter the plasma proteome profile of Atlantic salmon. These deviations can be helpful indicators for fish health and, therefore potential tools in the diagnosis of fish diseases. The present work aimed to identify local intestinal tissue responses and changes in plasma protein profiles of Atlantic salmon fed inactive dry *Candida utilis* yeast biomass, soybean meal, or combination of soybean meal based diet with various inclusion levels of *Candida utilis*. A fishmeal based diet was used as control diet. Inclusion of *Candida utilis* yeast to a fishmeal based diet did not alter the mor-phology, immune cell population or gene expression of the distal intestine. Lower levels of *Candida utilis* combined with soybean meal modulated immune cell populations in the distal intestine and reduced the severity of soybean meal-induced enteritis, while higher inclusion levels of *Candida utilis* were less effective. Changes in the plasma proteomic profile revealed differences between the diets but did not indicate any specific proteins that could be a marker for health or disease. The results suggest that *Candida utilis* does not alter intestinal mor-phology or induce major changes in plasma proteome, and thus could be a high-quality alter-native protein source with potential functional properties in diets for Atlantic salmon.

## Introduction

The composition of feeds used in salmon aquaculture has undergone significant changes over the last decades. The rapid growth in the aquaculture industry, but stable production of the

---

**Data Availability Statement:** All relevant data are within the manuscript and its Supporting Information files. The mass spectrometry

proteomics data have been deposited to the ProteomeXchange Consortium via the PRIDE partner repository with the dataset identifier PXD012051.

**Funding:** This study was funded by Foods of Norway, a Centre for Research-based Innovation (the Research Council of Norway; grant no. 237841/030) and and by BIOFEED - Novel salmon feed by integrated bioprocessing of non-food biomass (the Research Council of Norway; grant no. 239003).

**Competing interests:** The authors have declared that no competing interests exist.

major protein resource, fishmeal (FM), has led to increasing use of alternatives. Alternative protein sources are required to contribute to a well-balanced diet and to support optimal fish growth performance, health and disease resistance. Currently, proteins derived from insects [1], terrestrial animal co-products [2, 3] and microbial ingredients [4] are considered to be valuable alternatives. In commercial diets, plant-derived proteins have already replaced two third of the proteins of marine origin [5].

Plant ingredients are the most attractive protein sources due to their low cost of production, high protein content and availability [6]. Inclusion of plant ingredients in salmonid feeds can, however, result in reduced growth performance and feed utilization, and health issues due to anti-nutritional factors (ANF) [6, 7]. ANFs in plant-based diets have been associated with detrimental effects on the intestine of several salmonid species [7], and soybean meal-induced enteritis (SBMIE) is a well described condition in Atlantic salmon (*Salmo salar* L.) [8, 9]. Currently, in commercial salmon diets, the refined soy product with a reduced level of ANF, soy protein concentrate, is the primary protein source of plant origin [5] and has not been shown to cause pathological changes in the distal intestine (DI) of salmonids after short-term dietary exposure [9]. However, a certain degree of inflammation and ectopic epithelial cells have been observed in the DI of salmonids when fed diets based on FM and soy protein concentrate over a longer period of time [10].

Microbial ingredients have proven to be high-quality protein sources with the ability to mitigate the negative effects of plant-derived proteins [11]. Microbial ingredients such as yeast and bacteria contain bioactive compounds with immune-modulating properties that improve the changes caused by SBM [12–14]. Moreover, bacterial meal has been shown to prevent the development of SBMIE in a dose-dependent manner [15]. Mannan oligosaccharides, compounds found in yeast cell walls, have been used as a prebiotic and shown to protect the intestinal mucosa and reduce inflammation and leukocyte infiltration [16]. However, the degree of bioactivity of these compounds depends on the microbial origin as well as the fermentation conditions and downstream processing of the microbial product before incorporation into the salmon diet [11, 17].

Assessing the bioactivity of novel dietary microbial ingredients has traditionally involved the evaluation of local intestinal tissue responses, such as changes in morphology, gene expression or microbiome. Local intestinal responses can induce systemic responses that could contain new biomarkers for health and disease [18]. The innate immune system may respond to local inflammatory or immunomodulatory stimuli in the intestine of fish and in turn elicit changes systemically. The release of cytokines into the circulation stimulates hepatocytes to produce proteins and release them into the circulation to regain homeostasis [19]. Plasma proteomic analysis is a post-genomic tool that allows the investigation of complex biological systems involved in physiology and pathology. Plasma proteome profiles in response to certain inflammatory or immunomodulatory stimuli can be useful diagnostic tools for fish diseases and indicators of fish health.

In this study, an inactive dry yeast strain of *Candida utilis* (*C. utilis*) was used as an alternative protein source with functional properties in FM and SBM based diets. SBM was used as a dietary challenge to induce SBMIE. Increasing levels of *C. utilis* were included in the diets to evaluate the immunomodulating properties of the yeast, in particular, the ability to prevent and counteract SBMIE. We combine histomorphological evaluation, immunohistochemistry, morphometry and gene expression of the DI with plasma proteome analysis. By combining these methods, we aim to identify local intestinal tissue responses and changes in plasma protein profiles in Atlantic salmon resulting from dietary treatments. Our results show that inclusion of *C. utilis* as an alternative protein source does not alter the local morphology and

immune cell population in the DI or induce major changes in the plasma proteins of Atlantic salmon.

## Materials and methods

### Experimental ingredient and diet preparation

The inactive dry *C. utilis* yeast corresponded to a commercial product called Lakes States® Type B produced by LALLEMAND SAS (Blagnac, France). S1 Table shows the proximate composition of the test inactive dry yeast. The crude protein content was determined by multiplying nitrogen content by a conversion factor of 6.25. All diets used in this study were formulated to meet or exceeded the nutrient requirements of Atlantic salmon [20] (Table 1),

**Table 1. Ingredient and proximate chemical composition (g/kg) of control (FM) and experimental diets.** FM = Fishmeal; SBM = soybean meal; SBM25CU = soybean meal + 25 g/kg *C. utilis*; SBM50CU = soybean meal + 50 g/kg *C. utilis*; SBM100CU = soybean meal + 100 g/kg *C. utilis*; SBM200CU = soybean meal + 200 g/kg *C. utilis*; FM200CU = fishmeal + 200 g/kg *C. utilis*.

| Ingredient (g/kg) | Experimental diets | | | | | | |
|---|---|---|---|---|---|---|---|
| | FM | SBM | SBM25CU | SBM50CU | SBM100CU | SBM200CU | FM200CU |
| Fishmeal[a] | 425.4 | 193 | 193 | 190 | 190 | 190 | 269 |
| Soybean meal[b] | | 200 | 200 | 200 | 200 | 200 | |
| *Candida utilis*[c] | | | 25 | 50 | 100 | 200 | 200 |
| Wheat gluten | 150.6 | 150 | 139.4 | 115.2 | 93.5 | 72.9 | 118.8 |
| Corn gluten meal | | 60 | 60 | 60 | 60 | | 36.8 |
| Wheat flower | 168.6 | 181 | 167.4 | 169.6 | 142.9 | 132.3 | 166 |
| Fish oil[d] | 240 | 186 | 186 | 186 | 186 | 179.2 | 186 |
| Choline chloride[e] | 1.5 | 1.5 | 1.5 | 1.5 | 1.5 | 1.5 | 1.5 |
| MCP[f] | 6.2 | 10.8 | 10.8 | 10.9 | 10.9 | 10.8 | 9.6 |
| L-Threonine[g] | 0.9 | 2.4 | 2.2 | 2.2 | 1.7 | 1.2 | 1.0 |
| Premix Fish[h] | 6.3 | 6.3 | 6.3 | 6.3 | 6.3 | 6.3 | 6.3 |
| Rhodimet NP99[i] | | 2 | 2 | 2.2 | 2.2 | 2.6 | 1.5 |
| L-Lysine monohydrochloride[j] | 0.5 | 7 | 6.4 | 6.1 | 5 | 3.2 | 3.5 |
| Proximate chemical composition (g/kg) | | | | | | | |
| Dry matter | 936 | 942 | 936 | 943 | 956 | 937 | 932 |
| Crude protein | 445 | 409.1 | 401.9 | 397.5 | 400.3 | 383 | 394.4 |
| Crude lipid | 24 | 22.4 | 22.4 | 22.4 | 22.7 | 22.2 | 22.4 |
| Starch | 253.2 | 182.0 | 181.7 | 175.6 | 186.5 | 177.5 | 187.1 |
| Gross energy MJ/kg | 90.5 | 63.4 | 64.7 | 67.2 | 69.3 | 72.3 | 73.1 |
| Total ash | 14.0 | 8.9 | 9.3 | 9.2 | 8.8 | 9.0 | 10.2 |
| Phosphorus | 113.2 | 143.8 | 140.3 | 144.3 | 130.6 | 116.2 | 138.1 |

[a] LT fishmeal, Norsildmel, Egersund, Norway.

[b] Soybean meal, Non-GMO, Denofa AS, Fredrikstad, Norway.

[c] Lake States® Torula, Lallemand, USA.

[d] NorSalmOil, Norsildmel, Egersund, Norway.

[e] Choline chloride, 70% Vegetable, Indukern s.a., Spain.

[f] Monocalcium phosphate (MCP), Bolifor® MCP-F, Oslo, Norway Yara,

[g] L-Threonine, CJ Biotech CO., Shenyang, China.

[h] Premix fish, Norsk Mineralnæring AS, Hønefoss, Norway. Per kg feed; Retinol 3150.0 IU, Cholecalciferol 1890.0 IU, α-tocopherol SD 250 mg, Menadione 12.6 mg, Thiamin 18.9 mg, Riboflavin 31.5 mg, d-Ca-Pantothenate 37.8 mg, Niacin 94.5 mg, Biotin 0.315 mg, Cyanocobalamin 0.025 mg, Folic acid 6.3 mg, Pyridoxine 37.8 mg, Ascorbate monophosphate 157.5 g, Cu: CuSulfate 5H$_2$O 6.3 mg, Zn: ZnSulfate 151.2 mg, Mn: Mn(II)Sulfate 18.9 mg, I: K-Iodide 3.78 mg, Ca 1.4 g.

[i] Rhodimet NP99, Adisseo ASA, Antony, France.

[j] L-Lysine CJ Biotech CO., Shenyang, China.

produced by extrusion technology at the Center for Feed Technology (FôrTek) at the Norwegian University of Life Sciences (Aas, Norway), and stored at -20˚C prior to feeding. The extruded pellets were dried to ~ 6% moisture content before vacuum coating with fish oil. The diets consisted of a FM-based control diet (FM group) and the following six experimental diets; a diet containing 200 g/kg *C. utilis* (FM200CU group), and five diets containing 200 g/kg SBM together with 0 (SBM group), 25, 50, 100 or 200 g/kg *C. utilis* (SBM25CU, SBM50CU, SBM100CU and SBM200CU groups, respectively).

## Fish husbandry and feeding trial

The experiment was conducted according to laws and regulations for experiments on live animals in EU (Directive 2010/637EU) and Norway (FOR-2015-06-18-761). Vaccinated salmon (Aquavac PD7, MSD Animal Health, Bergen, Norway) were acquired from Sørsmolt AS (Sannidal, Norway) and maintained according to the guidelines established by the Norwegian Animal Research Authority at the Research Station Solbergstrand of Norwegian Institute of Water Research (Drøbak, Norway). Fish were acclimated to seawater, housed in 300 L tanks supplied with ultraviolet light treated seawater (8 ˚C; 34.5 g/L NaCl) in a 7–8 L per min flow-through system, and fed with a commercial marine-based compound feed not containing soybean-derived products (3-mm pellet; Polarfeed AS, Europharma, Leknes, Norway) under continuous light during a 4-month period prior to conducting the feeding trial. The water temperature, dissolved oxygen and pH level were measured and recorded daily. At the beginning of the experiment, 360 fish (average initial body weight of 526 g) were randomly assigned to 18 tanks (20 fish/tank) and acclimated to the FM based control diet for two weeks prior to feeding experimental diets. Feeding was approximately 20% in excess twice daily using automatic belt feeders based on a daily estimate of fish biomass and uneaten feed per tank, which was collected from the tank outlet after each feeding period. Following the acclimation period, each experimental diet was randomly allocated to the fish tanks (two tanks/diet) and fed for 30 days (period 1) as described above. After 30 days, the feeding strategy was changed, and new diets were fed for 7 days (period 2). As a control, one fish group received FM through the experiment. To assess whether *C. utilis* were able to counteract enteritis induced by SBM, four fish groups received SBM diets combined with different inclusions levels of CU (i.e. SBM25CU, SBM50CU, SBM100CU, SBM200CU) in period 1. One fish group received FM200CU in period 1 to evaluate if *C. utilis* in combination with FM alone would affect the DI. In period 2, the ability of *C. utilis* to prevent SBMIE was assessed as the diet was changed to SBM in this group. Finally, three fish groups were fed SBM diet to induce SBMIE in period 1, and in period 2 the diets were changed to either FM, FM200CU and SBM200CU to evaluate if these diets were able to reverse SBMIE. The feeding strategy is illustrated in Fig 1.

## Sample collection

At each sampling point (0, 7, 30 and 37 days), 8 fish per diet (4 fish per tank) were randomly selected and anesthetized by immersion in 60 mg/l of tricaine methanesulfonate (MS-222, Sigma-Aldrich, MO, USA) and subsequently euthanized by a sharp blow to the head. Before dissecting fish, fish weights and lengths were recorded, and blood samples were taken from the *vena caudalis* (tail vein) using a heparinized syringe and centrifuged (1300–2000 x g for 10 min) to isolate blood plasma, which was aliquoted and stored at –80˚C until proteomic analysis was performed. DI tissue was sampled and kept in RNAlater® (Merck KGaA, Darmstadt, Germany) at 4˚C for 24 h, and then at –80˚C, until RNA extraction. DI tissue samples were also collected and preserved for subsequent histology, morphometry and immunohistochemistry.

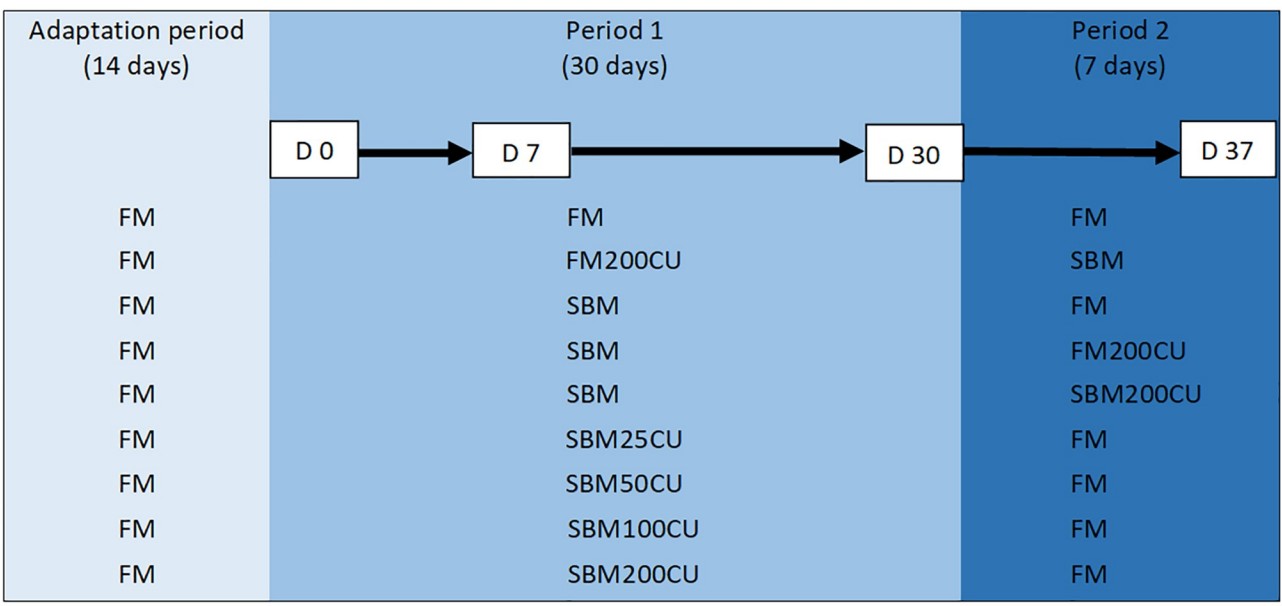

**Fig 1. Experimental design.** The adaptation period of 14 days was followed by experimental period 1 that lasted for 30 days. In experimental period 2, there was a shift in diets and this period lasted for 7 days. Sampling points are day 0, 7, 30 and 37. FM = Fishmeal; SBM = soybean meal; SBM25CU = soybean meal + 25 g/kg *C. utilis*; SBM50CU = soybean meal + 50 g/kg *C. utilis*; SBM100CU = soybean meal + 100 g/kg *C. utilis*; SBM200CU = soybean meal + 200 g/kg *C. utilis*; FM200CU = fishmeal + 200 g/kg *C. utilis*.

### Histology, immunohistochemistry and morphometric measurements

**Histology.** Approximately 1 cm segment of the DI was open longitudinally and the intestinal content was carefully removed. The tissue was fixed in 10% formalin for 48 h at room temperature and further processed according to routine histological procedures. Briefly, tissue was embedded in paraffin with an orientation to ensure longitudinal sectioning. Sections (2 μm) of paraffin-embedded DI tissue were mounted on glass slides (Menzel Gläser, Thermo Fisher Scientific, Braunschweig, Germany) and processed for staining with hematoxylin and eosin. A blinded, semi-quantitative histological scoring of the DI was performed using the criteria described in detail by Baeverfjord and Krogdahl [8]. Briefly, the criteria were: 1) shortening of both the simple and complex intestinal mucosal folds, 2) appearance of the enterocytes including supranuclear vacuolization, cellular heights and presence of intraepithelial lymphocytes (IELs), 3) widening of the central lamina propria of the simple and complex folds by connective tissue and 4) infiltration of leucocytes in the lamina propria. Each criterion was given a score ranging from 0 to 2, and half scores were included (i.e. 0, 0.5, 1, 1.5, 2) [13]. Score 0 indicated normal morphology and score 2 represented marked changes. Score 0.5 was regarded as changes within the normal range.

**Immunohistochemistry.** Histological sections of DI from the fish sampled at day 30 (8 fish/diet), prepared as described above, were subjected for immunohistochemical analysis, and the following diet groups were included: FM, FM200CU, SBM, SBM25CU and SBM200CU. CD3ε and CD8α positive T lymphocytes were identified in DI tissue sections by immunohistochemistry using a monoclonal anti-CD3ε antibody (dilution 1:600) [21] and a monoclonal anti-CD8α antibody (kindly supplied by Karsten Skjødt, dilution 1:50) [22], respectively. This analysis was performed as follows: formalin-fixed, paraffin-embedded DI sections (4 μm) were mounted on poly-L-lysine-coated glass slides (Superfrost Plus, Thermo Fisher Scientific, Braunschweig, Germany) and left to dry at 37˚C. The slides were incubated at 58˚C for 30 min

before deparaffinized in xylene and rehydrated in graded alcohols to distilled water. Antigen retrieval was done by using hydrated autoclaving at 121°C for 15 min in 0.01 M citrate buffer, pH6. Endogenous peroxidase was inhibited with 0.05% phenylhydrazine (0.05%; Sigma-Aldrich, MO, USA) in phosphate buffered saline, preheated to 37°C, for 40 min. The sections were stored in PBS overnight at 4°C and then incubated with normal goat serum (dilution 1:50; Sigma-Aldrich) in 5% bovine serum albumin /0.05 M tris-buffered saline to avoid non-specific binding for 20 min. The blocking solution was tapped off without washing, and the sections were incubated with primary antibody diluted in 1% bovine serum albumin/Tris-buffered saline for 1 h. Control sections were incubated with only 1% bovine serum albumin. The sections were incubated in the kit polymer-HRP anti-mouse (Dako En Vision+ System-HRP, Dako, Glostrup, Denmark), as a secondary antibody, for 30 min. Peroxidase activity was detected with 3,3'-diaminobenzidine following the kit procedure. The sections were counter-stained with hematoxylin for 30 s followed by washing in distilled water before mounting with Aquatex (Novoglas Labortechnik Langenbrinck, Bern, Germany). Unless otherwise stated, the sections were washed three times with phosphate buffered saline for 5 min between each step. All incubations took place in a humid chamber at room temperature.

**Morphometric measurements and calculation of immune cell density.** Morphometric measurements and calculation of the density of immune cells were performed from immuno-histochemically labelled sections mentioned above. ImageJ software, version v1.51r [23], was used to perform the measurements and calculations. Images were captured with a Zeiss Axio-cam 506 color camera connected to a light microscope (Zeiss Axio Imager M2, Carl Zeiss, Germany) at a 10 X magnification. The measurement scale was set to 2.26 pixels/μm in ImageJ and the measurements were converted from μm to mm. Counting of immunohistochemically labelled cells was performed using the multi-point tool. The freehand selection and segmented line selection were used to measure fold area and height, respectively. The fold height was mea-sured from stratum compactum to the tip of the epithelium lining the fold (S1 Fig). The fold area was measured from stratum compactum, including the middle of the fold base on each side, and the whole simple fold (S1 Fig). The immunohistochemically labelled cells were counted within the measured area of the simple fold. The density of CD3ε- or CD8α-labelled cells was calculated as follows: Cell density = (no. of labelled cells)/area. Simple folds were sub-jected to all the measurements mentioned above and the folds were selected as the first appro-priate simple fold located to the left of a complex fold. An appropriate fold was defined as a fold that appeared long, not bent and had an intact epithelium that was attached to the base-ment membrane all the way to the tip of the fold. Between 2–6 measurements were collected from each individual with a total of at least 30 measurements from each group for each mea-surement. A mean for each individual was calculated based on the measurements.

## RNA isolation, cDNA synthesis, quantitative PCR

A small piece of DI tissue (approximately 0.5 cm) from FM, FM200CU, SBM and SBM200CU diet groups (8 fish/diet) at day 30 were subject to gene expression analysis. Total RNA was extracted and purified using RNeasy® 96 kit (Qiagen, Valencia, USA) and QIAcube® HT sys-tem (Qiagen), according to the manufacturer's protocol. After the first washing step, on-col-umn DNase treatment was performed using PureLink TM DNase kit (Thermo Fisher Scientific, Waltham, Massachusetts). RNA concentration and quality were measured using NanoDrop TM 8000 spectrophotometer (Thermo Fisher Scientific). Purified total RNA was stored at −80°C until further analysis.

Prior to cDNA synthesis, all samples were normalized to 400 ng/μL. cDNA synthesis was performed using AffinityScript QPCR cDNA Synthesis kit following the manufacturer's

guidelines (Agilent Technologies, Santa Clara, CA, USA). The total reaction volume was 10 μL using 5 μL of Mastermix, 1.5 μL random hexamer primers, 0.5 μL AffinityScript RT/ RNase Block enzyme mixture and 3 μL DNase treated RNA. The resulting cDNA was stored at −80˚C before use.

All quantitative PCR (qPCR) reactions were performed in duplicates and conducted in 96 well plates on LightCycler® 480 system (Roche Diagnostics, Mannheim, Germany). Each reaction consisted of a total amount of 12 μL divided into 6 μL LightCycler 480 SYBR Green I Master (Roche Diagnostics), 2 μL primers (5μM), and 4 μL cDNA. The qPCR conditions were 95˚C for 5 min before a total of 45 cycles of 95˚C for 5 seconds, 60˚C for 15 seconds, and 72˚C for 15 seconds. To confirm amplification specificity, each PCR product was subject to melting curve analysis (95˚C 5 s; 65˚C 60 s; 97˚C continuously). Primers tested are listed in S2 Table. Glyceraldehyde-3-phosphate dehydrogenase (GAPDH) and hypoxanthine phosphoribosyl-transferase I (HPRTI) were chosen as reference for normalization since these genes did not show significant differential expression between the diets and have previously been described as suitable reference genes in the DI of salmon [24]. The crossing point (Cp) values were determined using the maximum second derivative method on the basis of the LightCycler® 480 software release 1.5.1.62 (Roche Diagnostics). The geometric mean of the CP-values for GDPH and HPRTI was used as an index. The qPCR relative expression of mRNA was calculated using the ΔΔCt method [25].

## In-solution digestion and protein sequence analysis by LC-MS/MS

Proteomic analysis was performed, according to methods described by Lagos *et al.* 2017 [26], using four biological replicates per treatment of plasma taken at the end of the first feeding period (30 days) (FM, SBM, SBM200CU and FM200CU). In brief, frozen plasma (−80˚C) was thawed and diluted to 40 μg of total protein in PBS, and the pH was adjusted to 8 by adding ammonium bicarbonate (Sigma-Aldrich, Darmstadt, Germany). Subsequently, the proteins were digested with 10 μg trypsin (Promega, sequencing grade) overnight at 37˚C. The digestion was stopped by adding 5 μL 50% formic acid and the generated peptides were purified using a ZipTip C18 (Millipore, Billerica, MA, USA) according to the manufacturer's instructions, and dried using a Speed Vac concentrator (Concentrator Plus, Eppendorf, Hamburg, Germany). The tryptic peptides were dissolved in 10 μL 0.1% formic acid/2% acetonitrile and 5 μL analyzed using an Ultimate 3000 RSLCnano-UHPLC system connected to a Q Exactive mass spectrometer (Thermo Fisher Scientific, Bremen, Germany) equipped with a nanoelectrospray ion source. For liquid chromatography separation, an Acclaim PepMap 100 column (C18, 2 μm beads, 100 Å, 75 μm inner diameter, 50 cm length) (Dionex, Sunnyvale CA, USA) was used. The mass spectrometer was operated in the data-dependent mode to automatically switch between MS and MS/MS acquisition. Survey full scan MS spectra (from m/z 400 to 2,000) were acquired with the resolution R = 70,000 at m/z 200, after accumulation to a target of 1e5. The maximum allowed ion accumulation times were 60 ms. The proteomic analysis was performed by the Proteomic core facility of the University of Oslo. The mass spectrometry proteomics data have been deposited to the ProteomeXchange Consortium via the PRIDE [27] partner repository with the dataset identifier PXD012051.

## Data analysis

**Histology, morphometric measurements, T-cell density and gene expression.** Non-parametric data from the histological evaluation were analyzed by Kruskal-Wallis followed by post hoc Dunn's test with a comparison of mean rank. Shapiro-Wilk normality test was used to test the normal distribution of the data from morphometric analyses and T-cell density and

further analyzed by one-way ANOVA followed by Tukey's multiple comparisons test. Morphometric analyses and T-cell density analyses were performed at the individual level using the mean of measurements of between 2–6 simple folds per fish. Results of qPCR (means ± standard deviations) were analyzed using One-way ANOVA with Dunnett's multiple comparison test ($a < 0.0001$). These analyses were performed in GraphPad Prism, version 7.00 and 8.0.1 (GraphPad Software Inc., San Diego, CA, USA).

**Proteomic data analysis.** The resulting proteomic data, MS raw files, were analyzed using MaxQuant and Perseus version 1.6.0.7 based on MS1 intensity quantification. Then identifications were filtered to achieve a protein false discovery rate (FDR) of 1%. Peptide identification was determined using fragment mass tolerance for the MS1 6 ppm and the fragment mass tolerance for the MS2 20 ppm. MS/MS spectra were searched against the salmon proteome available in September 2019 (https://www.uniprot.org/proteomes/query=taxonomy:82390), and reverse database searches were used in the estimation of FDR. The analysis was restricted to proteins reproducibly identified in at least two of the four replicates per diet, making the minimum number of peptides used to identify each protein an average value of 2. Row-wise normalization was applied to provide Gaussian-like distributions [28] for adjusting the differences among protein data. Protein raw data were transferred to log normalization; missing value imputation was used to replace the not identified proteins on the quantitative analysis and then performed on autoscaled data (mean-centered and divided by the standard deviation of each variable) [29]. A diagnostic plot was utilized to represent normalization procedures for normal distribution assessments [30]. Volcano plot analysis and data modeling were performed in R, using R package MetaboAnalystR [28]. UniprotKB database was used for the functional annotation of the proteins.

## Results

### Growth and general health

All groups of fish accepted their allocated diets and no significant differences were found in feed intake nor growth rate among dietary treatment. The average initial weight was 526 g and the average final weight was 667 g on day 37, considering that the weight was measured as bulk, this indicates that in general fish gained weight during the experimental period. The general health of the fish in this experiment was good, but two fish died during the experimental period, one from SBM50CU group and one from SBM100CU group, for unknown reasons.

### Histology

All fish sampled at day 0 showed normal DI morphology. Briefly, simple folds were long and slender with a thin lamina propria, whereas the complex folds were tall with a narrow lamina propria and a partial central core of smooth muscle. Intestinal epithelial cells were tall with the nucleus located basally, large vacuoles located apically and many IELs. Goblet cells were scattered among the epithelial cells towards the apex, and there was a higher presence of goblet cells at the apex of the complex folds. The lamina propria adjacent to the stratum compactum was thin and numerous eosinophilic granule cells were present.

At day 7 (Fig 2A), FM and FM200CU groups showed normal DI morphology as described above. In general, all fish groups fed diets containing SBM, including SBM diets with *C. utilis* inclusion, displayed changes in the DI morphology consistent with SBMIE (described in detail below). Nevertheless, in SBM25CU and SBM50CU, there was variation within the groups ranging from individuals showing no changes to other individuals with moderate changes in DI morphology.

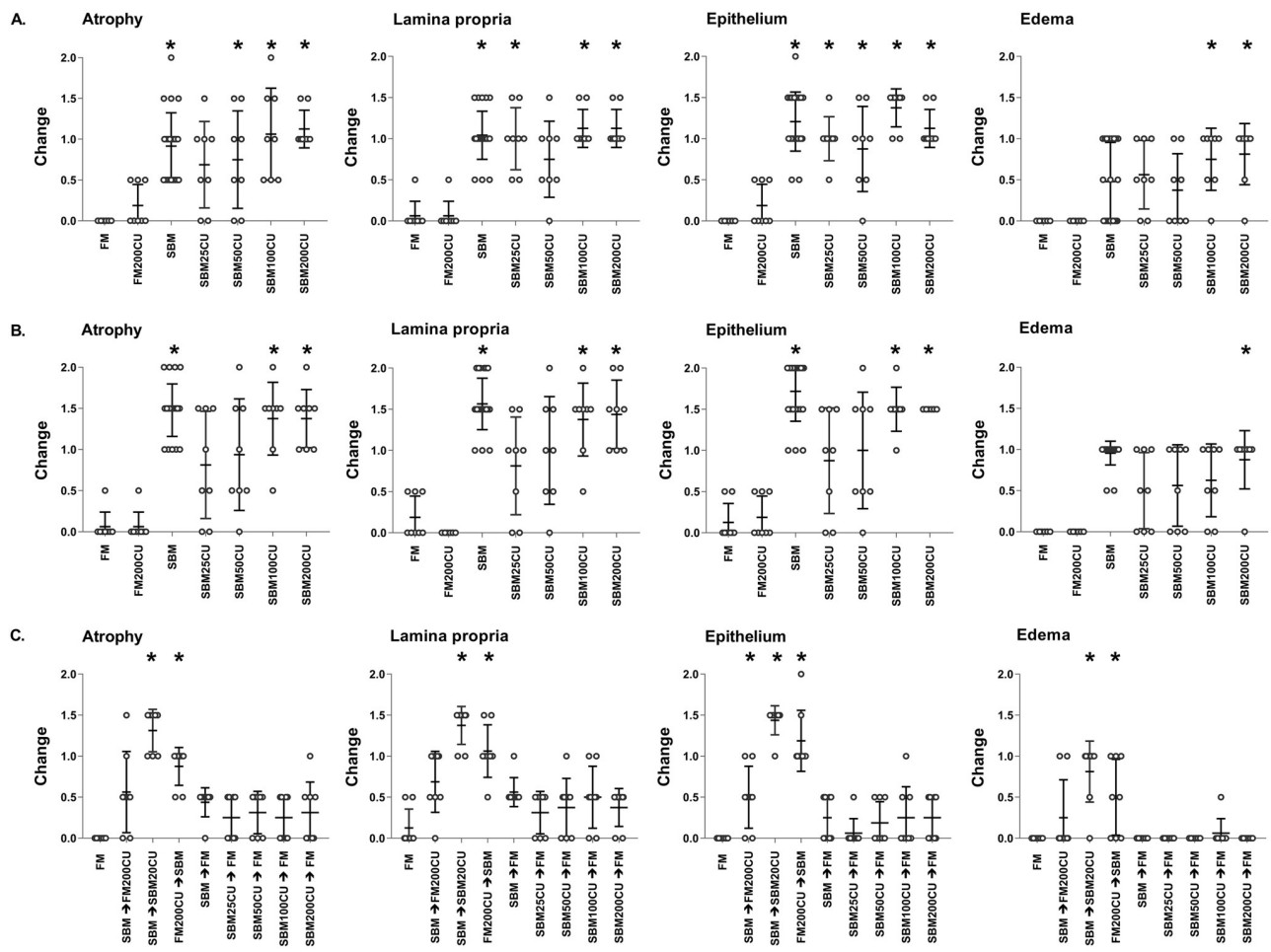

**Fig 2. Histological evaluation.** Histological evaluation of the distal intestine of Atlantic salmon based on atrophy, lamina propria, epithelium and edema at 7 (A), 30 (B) and 37 (C) days. Changes are scored from 0 to 2 where 0 indicates no changes and 2 indicates severe changes. Data are expressed as mean and standard deviation, n = 8 for all groups. Significant difference from the control fish fed FM based diet is denoted by an asterisk (*) ($p < 0.05$; Dunn´s test). FM = fishmeal; FM200CU = fishmeal combined with 200 g/kg *C. utilis* (CU); SBM = soybean meal; SBM25CU = soybean meal with 25 g/kg CU; SBM50CU = soybean meal with 50 g/kg CU; SBM100CU = soybean meal with 100 g/kg CU; SBM200CU = soybean meal with 200 g/kg CU.

At day 30 (Fig 2B), no morphological changes were seen in the DI of FM and FM200CU groups (S2A and S2B Fig). However, a moderate SBMIE was present in the DI of SBM (S2F Fig), SBM100CU and SBM200CU (S2E Fig) groups. Both simple and complex folds were shorter with a widening of the lamina propria within the folds and adjacent to the stratum compactum. A fusion of the simple folds was frequently observed. There was an increased presence of connective tissue in the lamina propria and the increased infiltration of leucocytes consisted mainly of eosinophilic granule cells and to a lesser extent of lymphocytes. The intestinal epithelial cells showed a reduction in height with nucleus displaced in a more apical position and small supranuclear vacuoles. In SBM25CU (S2C and S2D Fig) and SBM50CU groups, there was still a variation within the groups as seen on day 7.

At day 37 (Fig 2C), fish previously fed SBM, either alone or in combination with *C. utilis*, had normal DI morphology after being fed FM for 7 days. Similarly, most of the fish that had diets changed from SBM to FM200CU had DI morphology regarded as normal, but there were some fish in these groups that had mild enteritis. Changing diet from FM200CU to SBM

induced in general a mild SBMIE, whereas a shift from SBM to SBM200CU maintained a moderate SBMIE. The FM control group was normal, as described above.

No tank effects were observed at any of the sampling points. In Fig 2A and 2B, only fish from the tanks that were subjected to immunohistochemical analysis at day 30 are presented, i.e., two SBM-groups are omitted from the figure. Fish with diet shift from SBM combined with *C. utilis* to FM at day 37 had normal DI morphology but are excluded from Fig 2C as this group did not differ from the SBM group.

## Immunohistochemistry, morphometry, density of immune cells

At day 30, CD3ε positive cells showed an abundant presence at the base of the epithelium and extending along the entire length of simple folds of FM and FM200CU (Fig 3A and 3B) groups. Only a few CD3ε positive cells were observed in the lamina propria adjacent to the stratum compactum, and were rarely present in the lamina propria of the simple folds. A weak diffuse labelling was observed in the smooth muscle, which was interpreted as background labelling. The negative controls were blank.

The density of CD3ε positive cells in the simple folds of groups fed diets containing SBM was increased compared with the density in FM group. IEL's that showed labelling for CD3ε were located as individual cells at the base of the epithelium, but occasionally clusters of CD3ε-labelled IEL's were observed. CD3ε positive cells were more frequent in the lamina propria adjacent to the stratum compactum of fish fed SBM compared with fish fed FM (Fig 3D), but there were only a few CD3ε-positive cells present in the lamina propria of the simple folds (Fig 3C).

CD8α-labelled IEL's showed the same distribution as the CD3ε-labelled IEL's being located basally between the intestinal epithelial cells in all diets (Fig 3E–3G). In general, the presence of CD8α-labelled IEL's was lower than the presence of CD3ε-labelled IEL's.

Morphometric measurements of simple folds, both length and area, revealed no significant differences between FM and FM200CU groups. The simple folds in the DI of fish fed SBM, SBM25CU and SBM200CU were significantly shorter and had a considerably smaller area than the simple folds in fish fed FM200CU and FM. There were no statistically significant differences between the simple folds of fish fed diets containing SBM either alone or combined with *C. utilis* (Fig 4A and 4B).

The density of CD3ε and CD8α-labelled cells in fish of FM and FM200CU groups was significantly lower compared with the density in fish from groups fed diets containing SBM. There was a statistically significant difference between the density of CD8α-labelled cells in fish of SBM group and the density CD8α-labelled cells in fish from the SBM25CU (p = 0.0465) (Fig 4D).

## Gene expression

Among the tested genes, only mRNA expression of Aquaporin 8 (*aqp8*) was significantly down-regulated in the SBM and SBM200CU groups compared with the FM control group (Fig 5E). There was no significant difference between FM200CU and the FM control group. The transcription levels of superoxide dismutase 1 (*sod1*), glutathione S-transferase alpha 3 (*gsta3*), annexin A1 (*anxa*), and catalase (*cat*) were not different among dietary groups (p > 0.05) (Fig 5A–5D).

## Plasma proteome

We performed proteomic analysis on plasma sampled at day 30 from four individual fish from treatment FM, SBM, SBM200CU and FM200CU. In total, 286 salmon proteins were identified,

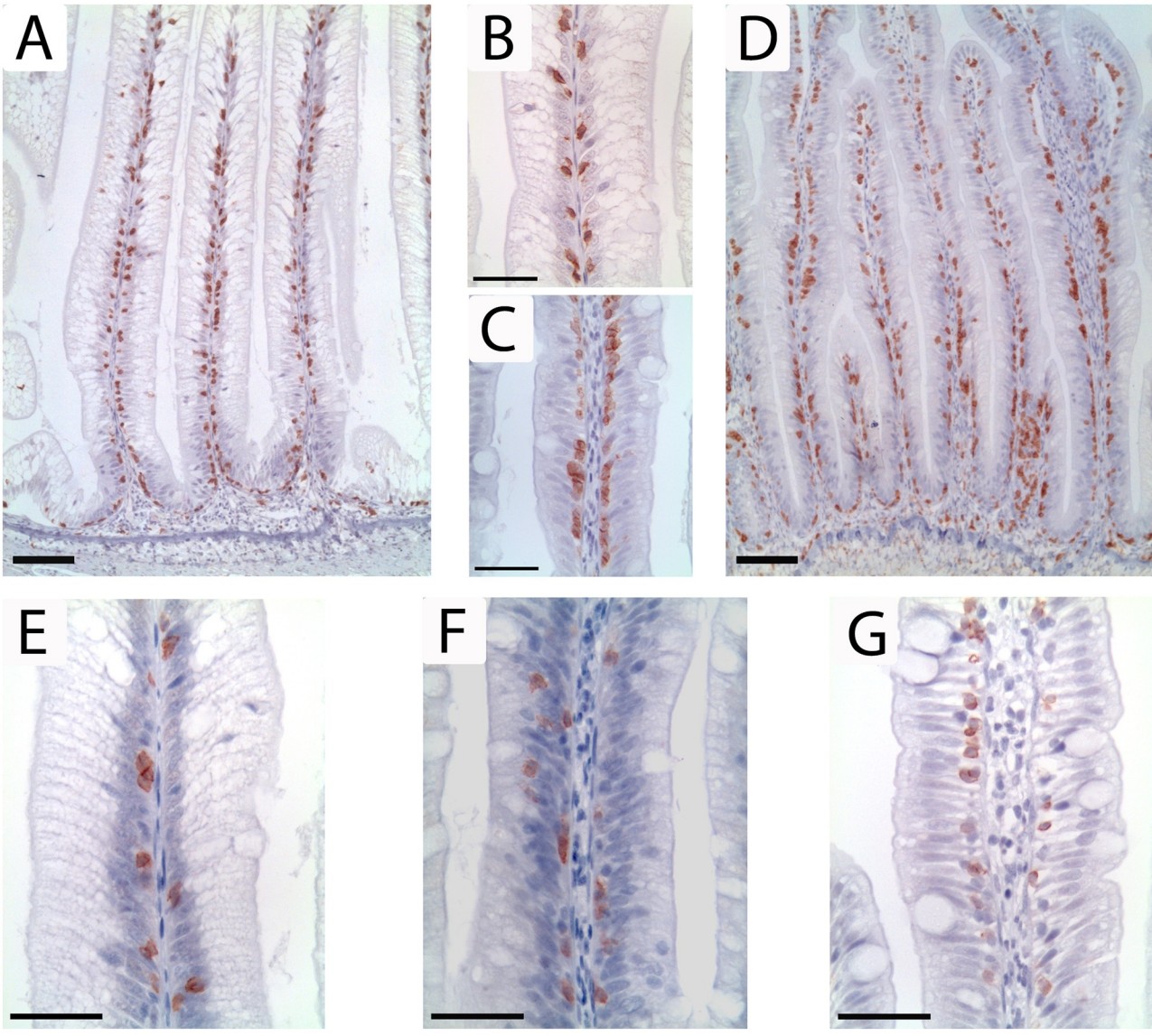

**Fig 3. Immunohistochemical staining for CD3ε and CD8α positive cells at day 30.** Immunohistochemical labelling (brown) showed an abundant presence of CD3ε positive cells at the base of the epithelium along the entire length of simple folds in all diet groups (A: FM200CU; D: SBM). At higher magnification, CD3ε positive cells were rarely present in the lamina propria of the simple folds in any of the diets (B: FM200CU; C: SBM). However, there was a higher number of CD3ε positive cells in the lamina propria adjacent to the stratum compactum in groups fed diets with SBM (D: SBM). CD8α positive cells were mainly found between the epithelial cells of all individuals of all diet groups (E: FM200CU; F: SBM200CU; G: SBM). Image A and D captured at 10x magnification, Image B, C, E, F and G captured at 40x magnification.

and after filtering for proteins present in at least two of the four individuals per diet, 158 proteins were selected for further analyses (S3 Table). This criterion was used due to the variability among fish. A Venn diagram was built up with the 158 proteins showing the overlapping of proteins detected in the four dietary treatments (Fig 6A). There were 126 plasma proteins shared between the four groups. Moreover, each dietary treatment presented unique proteins except for diet SBM200CU (D6) (Fig 6B). The five unique proteins present in D1 (FM) were creatine kinase, lymphocyte cytosolic protein 2, histone H3, kininogen-1 and electron transfer flavoprotein alpha polypeptide. While in D2 (SBM), we detected three unique proteins: fatty

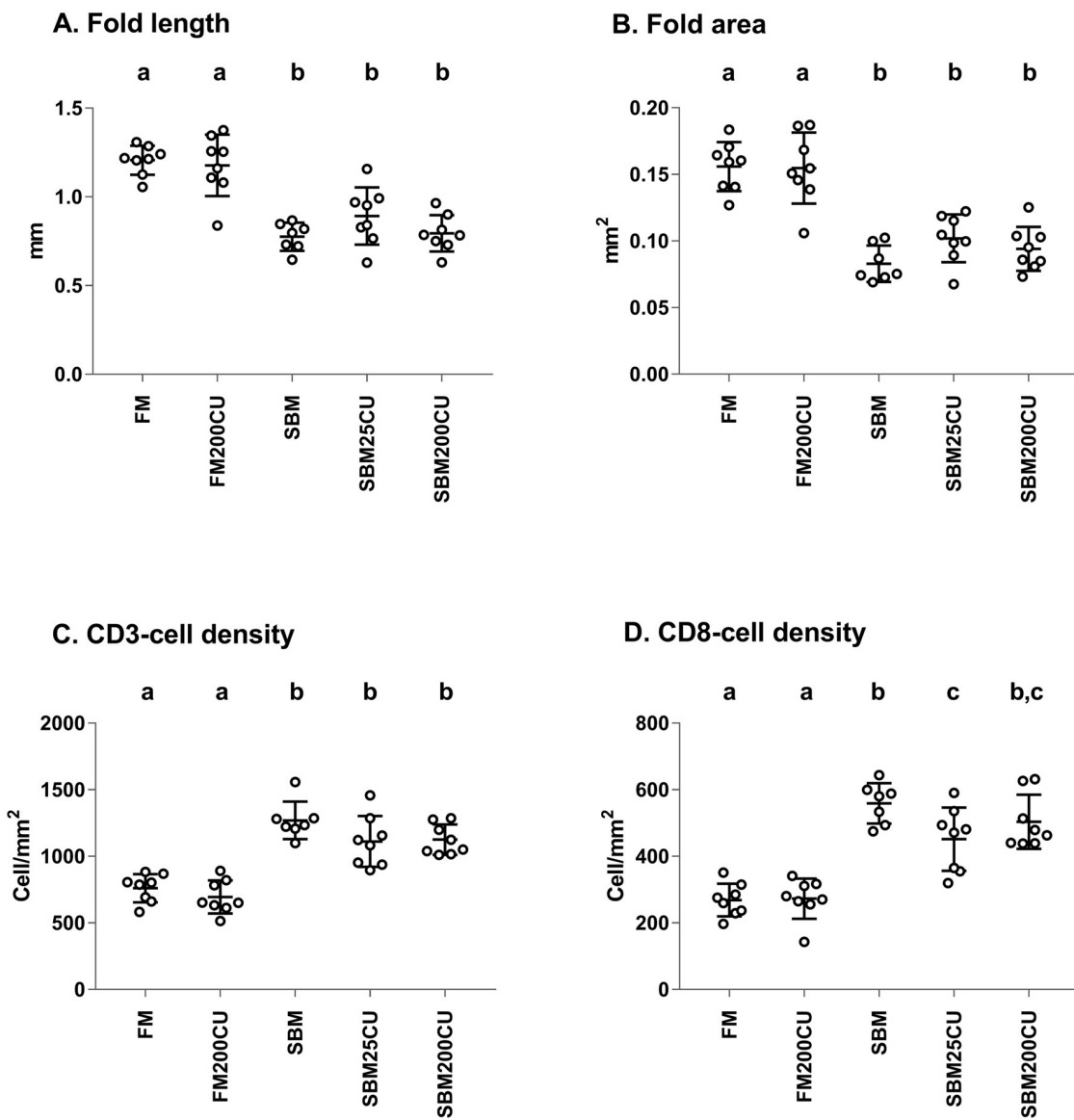

**Fig 4. Morphometry of simple folds in distal intestine at day 30.** Morphometric measurements of fold length (**A**) and fold area (**B**) of the simple folds of the distal intestine, and the density of CD3ε- and CD8α-positive T-cells in simple folds including the lamina propria adjacent to the stratum compactum (**C** and **D**). Data are expressed as mean for each individual ± standard deviation (SD), n = 7 for the SBM diet and n = 8 individuals per diet for the remaining groups. Groups with different letters on the upper x-axis are significantly different (p<0.05; Dunn's test).

acid-binding protein intestinal, transketolase and 14-3-3 beta/alph-1 protein. We did not detect unique proteins in D6 (SBM200CU), whereas in D7 (FM200CU) we found five unique proteins: GMP/IMP nucleotidase, hemoglobin subunit alpha, beta-globin, flavin reductase and L-lactate dehydrogenase.

In order to study the relative expression of proteins, we generated volcano plots comparing each dietary treatment to the control (S3 Fig). We observed that SBM induced the differential expression of nine proteins compared to the control diet (D1:FM), while diets containing yeast (200CU) show twelve and ten proteins respectively (SBM200CU, FM200CU) compared to the control diet (Table 2).

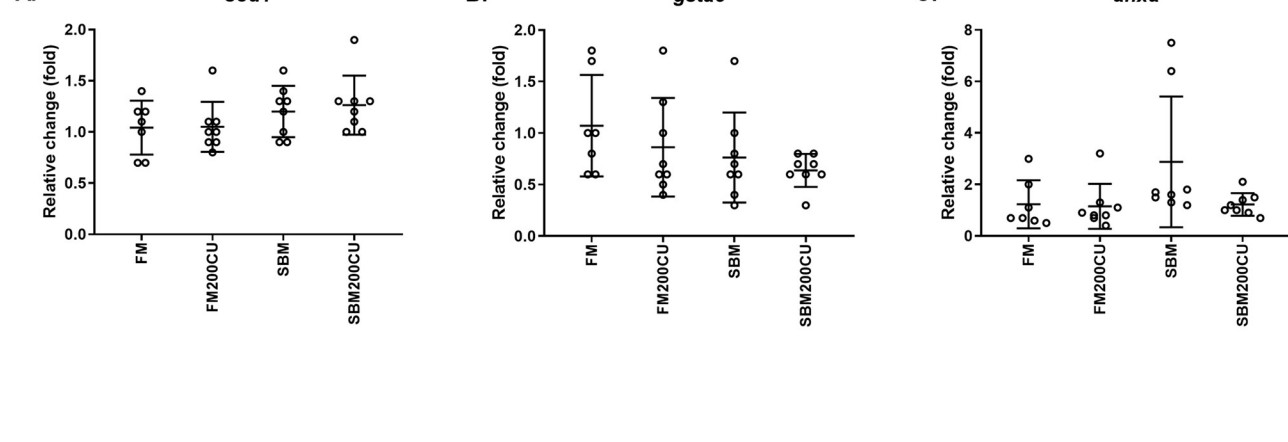

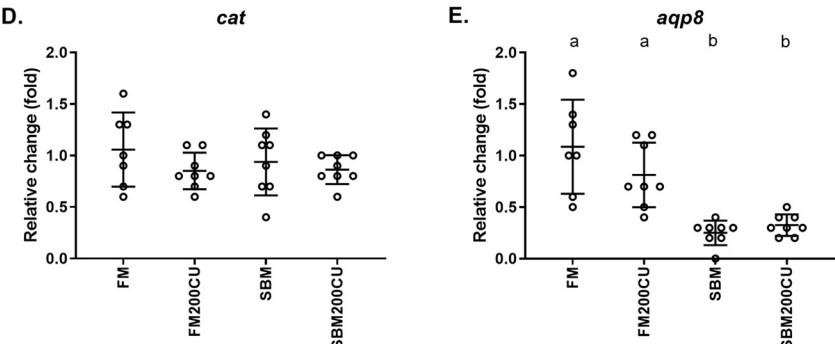

**Fig 5. Gene expression.** Quantitative PCR analyses of (**A**) superoxide dismutase 1 (*sod1*), (**B**) glutathione S-transferase alpha 3 (*gsta3*), (**C**) annexin (*anxa*), (**D**) catalase (*cat*) and (**E**) aquaporin 8 (*aqp8*) genes in the DI of Atlantic salmon fed a control fishmeal-based diet (FM), a diet containing 200 g/kg *Candida utilis* (FM200CU), and a diet containing 200 g/kg soybean meal (SBM) and one diet with 200 g/kg SBM in combination with 200 g/kg of *C. utilis* (SBM200CU) for 30 days. Data are mean −ΔΔCT ± SE (n = 7 for FM diet, n = 8 for the other groups). All relative fold changes are calculated in relation to the FM group.

## Discussion

Research on the effects of nutrition on fish health and disease has mainly focused on intestinal local immune responses rather than evaluating overall health impact. Therefore, the present study used an integrated approach to achieve a better understanding of the effect of feeding inactive dry *C. utilis* yeast, SBM and increasing levels of *C. utilis* yeast to Atlantic salmon in presence of SBMIE. Herein, we discuss how both local changes in the DI, including morphology, immune cell profile and gene expression, and systemic changes in the plasma proteome could reflect challenges posed by dietary treatments.

A previous study has shown that there were no significant negative effects on feed intake, specific growth rate or feed conversion ratio when up to 30% *C. utilis* was included in the diet for salmon [4], but DI morphology was not assessed. A study where parr where fed 200 g/kg *C. utilis* with FM showed that this diet combination did not alter the DI morphology when compared with FM control diet [31]. The present study is the first to demonstrate that FM200CU diet maintains a DI morphology similar to the FM based control diet in sea-water adapted farmed salmon. Furthermore, FM200CU presented a similar T-cell population profile in the DI compared with the control FM group, indicating no stimulation of the local T-cell

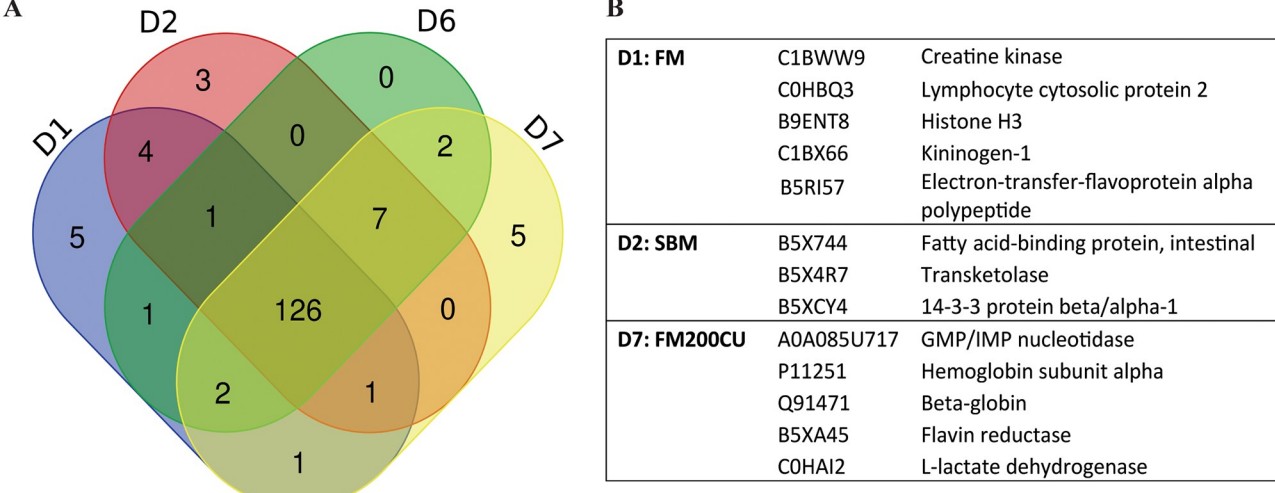

**Fig 6. Common and unique proteins expressed in plasma of salmon fed different diets.** (**A**) Venn-diagram showing the overlap between plasma protein sets detected across the four diet groups FM (D1), SBM (D2), SBM200CU (D6) and FM200CU (D7). (**B**) Unique proteins expressed in each dietary group.

population when SBMIE was not present. No significant alterations in mRNA expression further support the assumption that FM200CU does not impose any local effect in a normal DI.

On the other hand, it is widely known that SBM induces inflammatory changes in the DI morphology, and the local immune response of the DI to SBM has been described as a T-cell mediated inflammatory response [9, 32]. At d 30, the SBM, SBM25CU and SBM200CU groups had increased CD3ε and CD8α cell populations in the DI, which is consistent with this finding. The CD3ε- and CD8α-lymphocytes were mainly confined to the basal part of the DI epithelium with only a few CD3ε-labelled cells scattered in the lamina propria adjacent to stratum compactum. However, Bakke McKellep *et al.* [32] reported that lamina propria adjacent to stratum compactum and stroma of complex folds were rich in CD3ε-labelled cells in DI presenting with SBMIE. The CD3ε- and CD8α related observations might imply that the SBM used in these studies differed in immunostimulatory properties.

Interaction between *C. utilis* and SBM has been described by Grammes *et al.* where feeding 200 g/kg *C. utilis* together with SBM (200 g/ kg) prevented SBMIE development in salmon [13]. In our study, the severity of SBMIE in fish fed the highest inclusion levels of *C. utilis* (i.e. SBM100CU and SBM200CU) in combination with SBM was similar to that of fish fed the SBM diet. Adding lower levels of *C. utilis* to SBM diets (i.e. SBM25CU and SBM50CU) resulted in a large variation within the groups, ranging from normal morphology to moderate SBMIE. However, there was a decreased presence of CD8α-cell population in the SBM25CU group compared with the SBM group indicating that *C. utilis* has an immunomodulating effect locally in the DI when included at a low level in the SBM diet. Thus, partial prevention of SBMIE occurred only with lower inclusion levels of *C. utilis* which might differ with the previous work. This inconsistency could be due to differences in the degree of bioactivity of the *C. utilis* yeast, or in the ANF content of the SBM, and/or the experimental conditions. For example, Øverland and Skrede [11] have speculated that the inconsistent effect of yeast on host immunity can be attributed to yeast strain, fermentation conditions, and downstream processing when manufactured. Furthermore, Miadoková *et al.* concluded in their study that the biological activity of glucomannan isolated from *C. utilis* is dependent on the combined application with other biologically active compounds [33]. Also, the host itself can be the reason for

**Table 2. Significant proteins compared to control diet.** Relative expression of plasma proteins compared to the control, p-value threshold 0.05.

| FM v/s SBM | Protein name | Fold Change | log2(FC) | *p*-value |
|---|---|---|---|---|
| C0HAL2 | Elongation factor 1-alpha | 189.36 | 7.565 | 2.17E-13 |
| B5X5I8 | Profilin | 0.24669 | -2.0192 | 0.023289 |
| C0H808 | Tubulin beta chain | 4.0217 | 2.0078 | 0.02352 |
| B5DG39 | L-lactate dehydrogenase | 0.25567 | -1.9677 | 0.024172 |
| B5X1J1 | 72 kDa type IV collagenase precursor | 0.097995 | -3.3511 | 0.028933 |
| C0PU02 | Periostin | 0.27034 | -1.8872 | 0.043629 |
| B5DGM6 | Adenylate kinase isoenzyme 1 | 0.28582 | -1.8068 | 0.046312 |
| Q98SJ9 | Glycerol-3-phosphate dehydrogenase [NAD(+)] | 0.31016 | -1.6889 | 0.06348 |
| C1BX66 | Kininogen-1 | 104.54 | 6.7079 | 0.095789 |
| **FM v/s SBM200CU** | **Protein name** | **Fold Change** | **log2(FC)** | ***p*-value** |
| C0H808 | Tubulin beta chain | 17116 | 14.063 | 3.29E-15 |
| B5X1J1 | 72 kDa type IV collagenase precursor | 0.076752 | -3.7037 | 0.002394 |
| B5DG39 | L-lactate dehydrogenase | 0.24712 | -2.0167 | 0.02386 |
| B5RI57 | Electron-transfer-flavoprotein alpha polypeptide | 12192 | 13.574 | 0.023921 |
| B9ENN5 | Complement C1q-like protein 4 | 0.24833 | -2.0097 | 0.023926 |
| C0HAL2 | Elongation factor 1-alpha | 4.026 | 0.023934 | 0.023934 |
| C0HBQ3 | Lymphocyte cytosolic protein 2 | 13144 | 13.682 | 0.023995 |
| B5DG72 | Phosphoglucomutase-1 | 0.25169 | -1.9903 | 0.024093 |
| B5XGT3 | Lipocalin | 3.9621 | 1.9863 | 0.024132 |
| B5X0R1 | Complement component C7 precursor | 3.0761 | 1.6211 | 0.024173 |
| U5KQR2 | Heat shock protein 90-beta 2 | 3.2683 | 1.7086 | 0.034988 |
| C1BX66 | Kininogen-1 | 9560.9 | 13.223 | 0.03771 |
| **FM v/s FM200CU** | **Protein name** | **Fold Change** | **log2(FC)** | ***p*-value** |
| U5KQR2 | Heat shock protein 90-beta 2 | 144.16 | 7.1715 | 0.000177 |
| B5X1J1 | 72 kDa type IV collagenase precursor | 0.077739 | -3.6852 | 0.00245 |
| A0A085U717 | GMP/IMP nucleotidase | 0.24972 | -2.0016 | 0.02365 |
| B5RI57 | Electron-transfer-flavoprotein alpha polypeptide | 135.16 | 7.0785 | 0.023767 |
| C0HBQ3 | Lymphocyte cytosolic protein 2 | 145.5 | 7.1849 | 0.023813 |
| B5DG39 | L-lactate dehydrogenase | 0.25203 | -1.9883 | 0.023831 |
| C0HAI2 | L-lactate dehydrogenase | 0.0075714 | -7.0452 | 0.023863 |
| C0H808 | Tubulin beta chain | 3.9253 | 1.9728 | 0.024106 |
| B9ENC8 | Transgelin | 0.25797 | -1.9547 | 0.024391 |
| B5DGM6 | Adenylate kinase isoenzyme 1 | 0.27851 | -1.8442 | 0.045085 |

this inconsistency as it has been demonstrated that the severity of SBMIE can differ between strains of rainbow trout [34]. Additionally, the composition of the diet can induce variation in both the mucosa-associated and digesta-associated microbiota in Atlantic salmon [35, 36].

The mRNA expression profile of *aqp8* gene in the DI in the diet groups (FM, FM200CU, SBM, SBM200CU) indicates an association of *aqp8* with the resulting DI morphological and immune cell responses to nutritional challenges observed in this study. Our results confirm previous findings showing suppression of *aqp8* gene expression in intestinal inflammatory processes in salmon such as SBMIE [13, 37]. The relation between *aqp8* expression and inflammation is consistent with cell-based studies showing that reduced *aqp8* expression is linked to increased oxidative cell stress damage [38] and implying that *aqp8* is a key player in the maintenance of redox cellular status. Intestinal inflammation in salmon resulting from SBM has been shown to have wider systemic effects in plasma including increased insulin levels [39] and reduced plasma bile salt and cholesterol levels [40]. In the present study, we found three

proteins uniquely expressed in the SBM group; fatty acid-binding protein intestinal, transketolase and 14-3-3 protein beta. Salmonids fed SBM often present with reduced lipid digestibility [53], which can explain the presence of fatty acid-binding proteins in the plasma, due to tissue damage in the distal intestine.

The shift in diet at d 30 of the experiment showed that the developed enteritis observed in the DI of fish fed either SBM alone or in combination with *C. utilis* was resolved after feeding FM diet for 7 d. It is also important to point out that the degree of SBMIE was reduced from moderate to mild after 7 d feeding FM200CU diet to those fish fed SBM diet for 30 d. It can be suggested that all fish would have returned to a normal state if the experiment had lasted longer. Additionally, feeding FM200CU for 30 days prior to a diet shift to SBM did not prevent development of enteritis in the DI.

The present study revealed that dietary challenge induced the differential expression of specific proteins of the proteome but did not demonstrate a direct link to the local changes in the DI. This might have several explanations, such as sampling point, high variability among individuals and difficulty to measure individual feed intake. It should be noted that there is a lack of ideal biomarkers in gastrointestinal diseases in humans [18], such as inflammatory bowel diseases (IBDs), which further elucidates the difficulties of finding changes in the plasma proteome that can be directly linked to local inflammation or immunomodulation of the intestine. Nevertheless, there were specific proteins that were significantly expressed in our study that are of interest. We found that in the SBM group, there is an increased expression of periostin when compared with FM group. Periostin is a matricellular protein, belonging to the fasciclin family, that are also defined as an extracellular matrix protein that binds to cell-surface receptors [41]. This protein is an emerging biomarker reflecting type 2 inflammation in allergic diseases in humans [42], and it has also been suggested that periostin mediates intestinal inflammation in mice [43] and promote tumorigenesis in humans [44]. In both SBM and SBM200CU diet the expression of kininogen-1 is decreased. The plasma kallikrein-kinin system (KKS) has been shown to be activated in patients suffering with active IBDs where the high-molecular-weight kininogen and prekallikrein were significantly decreased in plasma assuming an increased KKS activation/consumption [45, 46]. Lipocalin is produced in the intestine by intestinal epithelial cells and is upregulated in presence of intestinal inflammation, and therefore is an attractive a biomarker for IBDs in humans [47, 48]. In our study, lipocalin was significantly decreased in plasma of the SBM200CU group. Additionally, L-lactate dehydrogenase was found to be differentially expressed in all dietary treatments, which would suggest that further experiments investigating the activity of this protein could be useful to elucidate the mechanism of action of yeast in salmon. L-lactate dehydrogenase, which catalyzes the conversion of lactate to pyruvate, consequently its activity increases under periods of muscular damage. Therefore, some authors suggest that the levels of lactate dehydrogenase in blood can be used as a diagnostic tool to predict cardiomyopathy syndrome or skeletal muscle inflammation on salmon, however the plasma levels do not correlate consistently with histological scores [49].

Differences between the plasma protein profiles of fish receiving SBM diets (SBM and SBM200CU) and FM diet were expected, as systemic changes may accompany a local tissue inflammation, thus plasma protein profiling could be a useful tool to determine the association between systemic response and outcomes of nutritional local challenges. However, the overall changes in the plasma proteome may be due to specific components in the diets (e.g. immunomodulating compounds of *C. utilis*) rather than local responses towards inflammation. Although obvious differences between SBM and SBM200CU were not manifested at morphological levels, we show that the SBM200CU group had a significant differential expression of complement C1q-like protein 4 and complement component C7 precursor in the plasma

when compared with control group (shown in Table 2). Complement factors are part of the innate immune system that enhance the ability of antibodies and phagocytic cells to clear microbes and damaged cells and promote inflammation. C7 is a precursor protein that, together with four other proteins, forms the membrane attack complex (MAC) of complement [50]. Complement C1q-like protein 4 has a C1q-like domain, similar to domains in C1 complex, and is assumed to take part in controlling aspects of inflammation, adaptive immunity and energy homeostasis [51]. Non-specific cellular and non-cellular immune responses have been reported when microbial ingredients have been included in fish diets [52, 53], and increase in complement activation in serum has been demonstrated in sea bass, jian carp and rainbow trout fed diets containing β-glucans [54–57]. Also, increased activities of complement in plasma of Atlantic salmon receiving an intraperitoneal injection of glucans from *Saccharomyces cerevisiae* have been demonstrated [58]. Interestingly, complement proteins were not differentially expressed in SBM or FM200CU groups, which suggests that the combination of SBM and yeast might trigger a different immune response than yeast alone. Additionally, heat shock protein 90, which is involved in the response to stress [59], was differentially expressed in the diets containing yeast (SBM200CU and FM200CU), but not in the group fed SBM alone. Currently, the effect of heat shock proteins on fish health is not clear, however our results showed that the inclusion of yeast in diets reduces its expression, which might be due to the immunomodulatory components present in the membrane of yeast.

This study combined traditional methods of assessing local tissue responses with plasma proteomic analysis in order to achieve a better understanding of the effects of nutrition on fish health and disease, and to identify systemic protein profiles in response to the dietary treatments. The inclusion of 200 g/kg of *C. utilis* yeast to a FM based diet (i.e. FM200CU) as a novel protein source with potential functional properties produced similar DI morphology, immune cell population and gene expression profiles, compared with a FM based control. Feeding the SBM diet induced SBMIE while feeding lower inclusion levels of *C. utilis* in combination with SBM (i.e. SBM25CU and SBM50CU) reduced the severity of SBMIE. Interestingly, higher inclusion levels of *C. utilis* yeast (i.e. SBM100CU and SBM200CU) did not show any significant protection against SBMIE, which is in contrast to a previous study that has shown protection. Changes in the plasma proteomics between groups with different dietary challenges were observed, however, their potential for the identification biomarkers for health or disease is yet to be elucidated. The absence of any clear alterations of expression of proteins within specific biological pathways would indicate that the effects due to diets were mainly restricted to local tissue responses.

In conclusion, our results suggest that C. utilis does not alter intestinal morphology or induce major changes in plasma proteome, and thus could be a high-quality alternative protein source with potential functional properties in diets for Atlantic salmon.

## Supporting information

**S1 Table. Chemical composition (g kg$^{-1}$) of dry *Candida utilis* biomass.**
(DOCX)

**S2 Table. Primers used in qPCR analysis.**
(DOCX)

**S3 Table. 158 proteins identified and their quantitative values.**
(XLS)

**S1 Fig. Morphometric measurement.** Red line indicates the measurement of fold height from the tip of the simple fold to the stratum compactum. The yellow line indicates the fold area

including the simple fold and the lamina propria adjacent to the stratum compactum.
(JPG)

**S2 Fig. Histology.** Representative histomorphological images from hematoxylin and eosin-stained sections of the distal intestine of Atlantic salmon fed control FM diet **(A)** and experimental diets **(B-F)**. Normal morphology was seen in FM **(A)** and FM200CU **(B)** groups. Moderate changes associated with SBMIE was observed in the distal intestine of salmon fed SBM (**C**: SBM25CU, **E**: SBM200CU, **F**: SBM). Low inclusion of *C. utilis* to the SBM diet showed variation within the group ranging from individuals showing little changes (**D**: SBM25CU) to individuals with moderate changes in DI morphology (**C**: SBM25CU). All images are captured at 4x magnification with a scale bar (100μm).
(TIF)

**S3 Fig. Volcano plot.** Volcano plots showing the relative expression of plasma proteins from fish fed SBM (A), SBM200CU (B) or FM200CU (C) compared to the control (FM). ANOVA plot with p-value threshold 0.05.
(TIF)

## Acknowledgments

The authors thank Aleksandra Bodura Göksu for helping with immunohistochemistry techniques and to Ricardo Tavares Benicio for helping with feed manufacture, feeding the fish and sampling.

## Author Contributions

**Conceptualization:** Felipe Eduardo Reveco-Urzua, Liv Torunn Mydland, Charles McLean Press, Leidy Lagos, Margareth Øverland.

**Data curation:** Felipe Eduardo Reveco-Urzua, Liv Torunn Mydland, Leidy Lagos.

**Formal analysis:** Felipe Eduardo Reveco-Urzua, Mette Hofossæter, Mallikarjuna Rao Kovi, Ragnhild Ånestad, Charles McLean Press, Leidy Lagos, Margareth Øverland.

**Funding acquisition:** Margareth Øverland.

**Investigation:** Felipe Eduardo Reveco-Urzua, Liv Torunn Mydland, Leidy Lagos.

**Methodology:** Felipe Eduardo Reveco-Urzua, Mette Hofossæter, Mallikarjuna Rao Kovi, Ragnhild Ånestad, Randi Sørby, Charles McLean Press, Leidy Lagos, Margareth Øverland.

**Project administration:** Margareth Øverland.

**Supervision:** Felipe Eduardo Reveco-Urzua, Liv Torunn Mydland, Randi Sørby, Charles McLean Press, Margareth Øverland.

**Validation:** Felipe Eduardo Reveco-Urzua, Mette Hofossæter, Mallikarjuna Rao Kovi, Ragnhild Ånestad, Randi Sørby, Charles McLean Press, Leidy Lagos.

**Visualization:** Felipe Eduardo Reveco-Urzua, Mette Hofossæter, Mallikarjuna Rao Kovi, Ragnhild Ånestad, Randi Sørby, Charles McLean Press, Leidy Lagos.

**Writing – original draft:** Felipe Eduardo Reveco-Urzua, Mette Hofossæter, Ragnhild Ånestad, Leidy Lagos, Margareth Øverland.

**Writing – review & editing:** Felipe Eduardo Reveco-Urzua, Mette Hofossæter, Mallikarjuna Rao Kovi, Liv Torunn Mydland, Ragnhild Ånestad, Randi Sørby, Charles McLean Press, Leidy Lagos, Margareth Øverland.

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
