## [Decision Letter · Decision Letter 0]

9 Jul 2019

PONE-D-19-15096

Candida utilis yeast as a functional protein source for Atlantic salmon (Salmo salar L.): Local intestinal tissue and plasma proteome responses

PLOS ONE

Dear Dr. Øverland,

Thank you for submitting your manuscript to PLOS ONE. After careful consideration, we feel that it has merit but does not fully meet PLOS ONE’s publication criteria as it currently stands. Therefore, we invite you to submit a revised version of the manuscript that addresses the points raised during the review process.

Most importantly, you should pay attention to the statistical analysis of the proteomic data, as carefully explained by Reviewer 3. In addition, both Reviewers 1 and 2 express concerns about the discussion of the manuscript, which should be carefully revised. 

We would appreciate receiving your revised manuscript by Aug 23 2019 11:59PM. To enhance the reproducibility of your results, we recommend that if applicable you deposit your laboratory protocols in protocols.io, where a protocol can be assigned its own identifier (DOI) such that it can be cited independently in the future. For instructions see: http://journals.plos.org/plosone/s/submission-guidelines#loc-laboratory-protocols

We look forward to receiving your revised manuscript.

Kind regards,

Annie Angers, Ph.D.

Academic Editor

PLOS ONE

Journal Requirements:

3. Please amend the manuscript submission data (via Edit Submission) to include author Leidy Lagos.

Reviewers' comments:

Reviewer's Responses to Questions

**Comments to the Author**

1. Is the manuscript technically sound, and do the data support the conclusions?

Reviewer #1: Partly

Reviewer #2: Yes

Reviewer #3: Partly

2. Has the statistical analysis been performed appropriately and rigorously? 

Reviewer #1: Yes

Reviewer #2: I Don't Know

Reviewer #3: Yes

3. Have the authors made all data underlying the findings in their manuscript fully available?

Reviewer #1: Yes

Reviewer #2: Yes

Reviewer #3: Yes

4. Is the manuscript presented in an intelligible fashion and written in standard English?

Reviewer #1: Yes

Reviewer #2: Yes

Reviewer #3: Yes

5. Review Comments to the Author

Reviewer #1: The present work studies how the addition of C. utilis in soybean meal based diets impacts or reverts the changes induced when compared to a control fishmeal based diet. The results show that low levels of yeast reduced the severity of the enteritis caused by soybean meal based diets and C. utilis is proposed as an alternative protein source in functional diets for Atlantic salmon. In my opinion, this is a very attractive and complete study that combines different techniques in order to evaluate Atlantic salmon intestinal health. However, I have some comments and concerns that I believe should be addressed before considering this manuscript for publication.

Major concerns:

The experimental design seems carefully planned and complex, with many groups and variables. However, not all groups were used for all analyses and there is no explanation why. For example, it is stated that low doses partially prevent SBMIE, but these groups were not included in the gene expression or proteomic analyses. In addition, an explanation of the different effect that low and high levels of C. utilis have on inflammation should be included.

The discussion seems to focus on randomly selected proteins. There is the description of the function of some of these proteins but I fail to see the point and relevance in this particular study. In my opinion, the whole discussion should be revised to follow a particular relevance point regarding this research, animal model and application.

Other comments:

L71: This sentence looks weird. Maybe the authors meant: “…an inactive dry Candida utilis strain was used…” or “… an inactive dry yeast strain of Candida utilis… “.

L108: Vaccinated against?

When mentioning average body weight values the error should be included. For instance, L116.

L123: The feeding strategy was changed to what? In general, the experimental design is not clearly explained and even though there is a nice explanatory figure the text should be more clear.

L200: The amount of tissue used for RNA isolation should be stated.

L215: The concentration of the primers should be stated. Also, in the supplementary figure of the primer sequences the symbol for GDPH is GADPH, please unify the nomenclature.

L253: I think a reference to the salmon proteome should appear here.

L269: Why a series of PCA? Were several analyses performed? If so, they should be described. Also, verify the use of analyses instead of analysis for the plural.

L289: The average initial and final weight are mentioned (again I’m missing the errors), but there is no comparison among groups. Were there differences? Maybe a table should be included with the group values?

L303: “described in detail below” (remove the “ed”)

L293: section Histology: A panel of pictures showing the most significant changes (like the ones described in L308-312) would be nice. Also, how do the authors know exactly which cell types are present (L311-312) without specific markers? Some of these cells are difficult to identify in intestinal sections.

L323-324: Why where those groups excluded from the figure?

L366-367: The difference mentioned here is not represented in the figure, in fact, regarding the figure statistics these groups were not different in this parameter.

L387: The sudden change in nomenclature does not simplify the reading of the data, it actually complicates it. The nomenclature should be maintained along the manuscript (including the figures). In addition, why were the groups with low yeast dose excluded from this analysis? This should be explained in detail, particularly because the work concludes that the low doses are the ones having more positive effects but there are no results about them in this part of the study.

L403 and Fig6C: This heatmap seems to show differential expression values. If this is correlation I don’t understand against what each molecule is correlated. Please revise these results and explanations.

L409: “Analyses were carried out first with….” Why first? Was there a second analysis?

L445: These inconsistencies should be further explained.

L473: Delete DI.

L473-475: The statements about the role of aqp8 in homoeostasis should be further supported.

L489-492: I do not see the point of mentioning these particular roles of MHCI here. Maybe the authors are trying to make a point I do not understand.

L494: One study or several? If one, change for “A previous study”. Otherwise use plural.

L509: Downregulation in which tissue?

Figure 3: Scale bars should be added to the pictures.

Figure 5: The fold change is relative to what?

Reviewer #2: Review of PONE-D-19-15096 ” Candida utilis yeast as a functional protein source for Atlantic salmon (Salmo salar L.): Local intestinal tissue and plasma proteome responses”.

This manuscript describes the effects of using a yeast strain as a functional protein source to boost gut health in fish at risk of intestinal enteritis when fed soy bean-based meal. The study tackles a highly relevant problem and is generally very well written and clear. Overall, this study will provide an important contribution to the field and fit well into the PLoS one journal. Below I provide some general and more specific comments to help the authors improve the manuscript for publication.

General comments

I want to applaud the authors for performing such a holistic analysis opposed to using just one single method. This is very much needed and sets a great standard for future studies.

However, what I miss a little is an attempt to better link all the data sets together in a biological way. Could simply be through discussing results, but in a way to e.g. link the genes tested for expression responses with the proteins analyses all the way to the tissue histological data. Maybe add a conceptual figure with the entire path from DNA -> RNA -> proteins -> tissues and the highlight where you have generated data and results. As is, the different data types are treated quite independently and in a somewhat random order.

For example, the discussion on lines 467-469 represents a good example of how gene expression data is integrated with insights from histology and immune response data.

Contrary, on lines 484-488 is an example where the protein plasma data is mainly discussed in isolation, whereas it would benefit from a stronger focus on the underling biological interactions with e.g. gene expression, histology and immune response data.

Specific comments

L. 108: What was the fish vaccinated against? Was it a standard vaccination program or inly for a specific disease? Pease specify.

L. 158: Which and how many fish were used for this immunohistochemistry analysis? Was it the same 8 fish as above or new ones? Please specify.

L. 224: Do you mean “… as index” instead of “… an index”?

L. 229 + 231 + throughout: Be consistent whether or not you add a space between the degree symbol and “C” when listing temperature values.

L. 253: Please provide a reference for the salmon proteome reference database.

L. 262-263: Can you clarify this description as it sounds like all individuals are already part of "a cluster" when you begin. So, when does the model know to end the analysis? Is it e.g. trying to minimise the number of clusters based on some preset value? This remains unclear as currently written.

L. 275: Add space before “[31]”.

L. 388-389: It is not clear to me how this "simplifies" except using fewer letters. The figures have plenty of space to write out the FM and SBM names respectively, and these are much more informative than having to cross read between the text here and the figure to know what D1-D4 represents. I would prefer to just keep the FM and SBM based names in the figures as well. Same for Fig. 7.

L. 438-444: Could this maybe also be due to individual responses among fish, and that these responses are partly controlled by the different genotypes among individuals? For example, if host genotype selects for different gut-microbiota communities, one could speculate that such different gut environments will differ in their ability to respond to the provided C. utilis strains? I agree this is very speculative, but the explanation do warrant mention as opposed to only focusing on how external parameters could explain the observed variation in gut responses; which seems to be the standard way of interpreting these types of gut data.

L. 503: re-write to "proteins with significant over expression in ... ". The proteins themselves are not significant, it is their expression level that is.

Figure 2: Please add labels for all Y axes.

Figure 3: Please clarify the different colors in the figure images. Are CD8 positive cells the purple colored ones? Or are both CD3 and CD8 colored brown? If so, what are the purple stains?

Figure 4: Discuss whether the low sample size (n=7) may cause type-2 errors for the non-significant comparisons?

Figure 5: Add both a and b letters between the two diet categories in the plots where there are significant differences.

Also, same concern as for figure 4 about regarding lack of statistical power with the relatively low sample sizes. Was any prior power analyses performed to inform the used sample sizes? There are some differences in e.g. plot C that could potentially be significant with more statistical power, which could for example be obtained through an increased sample size.

Reviewer #3: General Impressions:

In this manuscript, the authors performed a feeding trial on Atlantic salmon in which fish were fed fish meal or soybean meal-based diets that were supplemented with varying amounts of Canadis utilis yeast. Changes in intestinal morphology and the plasma proteome of the fish were monitored at several points in the study in order to monitor both intestinal and systemic effects of the different diets.

Overall, the experimental design appears valid and the data collection and analysis of the histological data is appropriate. However, in the eyes of this reviewer, the analysis of the plasma proteomics data could be improved to allow for more meaningful interpretation of the data. The current methodology does not provide significant evidence that the observed changes in the plasma proteome are related to the differences in diet. However, after the authors address the following comments, the manuscript would likely be suitable for publication.

Major Comments:

• The authors relied primarily on Spearman’s correlation coefficients to identify proteins of interest from the plasma proteomics data. Therefore, the authors are essentially looking for cases where protein abundances do or do not correlate across the different diets. However, since there is a clear control in the experiment (fish meal-based diet), it seems that comparing all protein abundance to that control would be more informative. Then, emphasis would be placed on proteins that have ‘abnormal’ expression relative to the fishmeal fed fish. This would also allow for volcano plots to be generated and included in the manuscript, which would aid reader understanding.

• Further description of how the quantitative values used in the proteomics analysis is needed. It appears that precursor abundance (from the MS1 scans) were used, but that is not explicitly stated. If that is the case, was peak intensity or peak area used? Also, was the ‘match between runs’ feature enabled in MaxQuant?

• A table showing all of the proteins identified in the plasma analyses and their quantitative values should be provided in the supplement. Ideally, fold change values should also be included. It’s possible that such a table was uploaded to PRIDE, but the PXD number provided does not bring up any results when searched. This is likely because the dataset has not been publicly released but the login info should be provided to reviewers.

• It is concerning that the fishmeal-based and soybean-based diets don’t cluster in either the heatmap or the PCA plots. It seems likely that such large differences in the feed would causes systemic changes in the blood which would lead to high correlation between the D1 and D7 samples, but that was not observed. This suggests that either the analysis method was flawed, or that individual variation was too large in order to adequately detect differences from the feed.

• Although this reviewer is not well versed in the VIP method used here, it seems that this technique is likely to put too much emphasis on proteins that are correlated purely by chance. Further justification of this technique should be included, or the analyses should be removed.

• It would be helpful if the reviewers included a few representative images of the changes in morphology observed between the groups that are summarized in Fig. 2. This could be included as an additional supplemental figure.

Minor Comments:

• L252: The fragment ion tolerance used here is much too large for orbitrap data. A tolerance of 0.02 Da would be more appropriate.

• L253: Where was the salmon proteome database derived from? Was it a public repository such as uniprot? If so, when was it downloaded?

• L316: No data is shown in this figure where SBM + C. utilis was fed first, followed by FM. This sentence says otherwise.

• Fig. 4 caption – L371: What is meant hear by ‘data are expressed as mean � SD for each individual’? Isn’t each point just representative of the mean for each fish? This should be clarified.

• L407-408: It doesn’t make sense to say that proteins ‘belong’ to a specific domain. Please revise.

• Fig. 7 caption – L424: What do the colors in panel C represent? Also, why are proteins with VIP scores < 2 included here?

• L473: Remove ‘DI.’

• L476: Change considering to considered.

6. PLOS authors have the option to publish the peer review history of their article (what does this mean?). If published, this will include your full peer review and any attached files.

Reviewer #1: No

Reviewer #2: No

Reviewer #3: No

---

## [Author Response · Author response to Decision Letter 0]

23 Aug 2019

Reviewer #1: The present work studies how the addition of C. utilis in soybean meal based diets impacts or reverts the changes induced when compared to a control fishmeal based diet. The results show that low levels of yeast reduced the severity of the enteritis caused by soybean meal based diets and C. utilis is proposed as an alternative protein source in functional diets for Atlantic salmon. In my opinion, this is a very attractive and complete study that combines different techniques in order to evaluate Atlantic salmon intestinal health. However, I have some comments and concerns that I believe should be addressed before considering this manuscript for publication.

Major concerns:

The experimental design seems carefully planned and complex, with many groups and variables. However, not all groups were used for all analyses and there is no explanation why. For example, it is stated that low doses partially prevent SBMIE, but these groups were not included in the gene expression or proteomic analyses. In addition, an explanation of the different effect that low and high levels of C. utilis have on inflammation should be included.

The discussion seems to focus on randomly selected proteins. There is the description of the function of some of these proteins but I fail to see the point and relevance in this particular study. In my opinion, the whole discussion should be revised to follow a particular relevance point regarding this research, animal model and application.

Unfortunately, we could not analyze all samples in all analyses. Based on earlier experience with microbial ingredients, we assumed that the highest concentratiosn of CU would have the most effect when added to SBM based diet. Thus e.g., in the proteomic analyses, only FM (D1), SBM (D2), SBM200CU (D6) and FM200CU (D7) groups were subjected to analyses. In other analyses, we have included higher number of samples/dietary treatments. 

We have revised and re-arranged large parts of the discussion.

Other comments:

L71: This sentence looks weird. Maybe the authors meant: “…an inactive dry Candida utilis strain was used…” or “… an inactive dry yeast strain of Candida utilis… “.

Changed to: “… an inactive dry yeast strain of Candida utilis…”

L108: Vaccinated against?

Farmed Atlantic salmon in Norway are routinely vaccinated as pre-smolts against a number of diseases. In this case, the Aquavac PD7 was used (contains vaccine against bacterial furunculosis, vibriosis, cold water vibriosis, winter ulcer) and viral (infectious pancreatic necrosis (IPN), pancreas disease, infectious salmon anaemia (ISA)) diseases.

We have added this, thus the sentence now reads: “Vaccinated salmon (Aquavac PD7, MSD Animal Health, Bergen, Norway) were acquired from…”.

When mentioning average body weight values the error should be included. For instance, L116.

To avoid too much stress for the fishes at the start of the experiment, it is common to register bulk weigh of all fish per tank. Thus, in this experiment, we have only the bulk starting weight of the 20 individuals per tank. We can therefore only give an average initial body weight (526 g) withour the error. 

L123: The feeding strategy was changed to what? In general, the experimental design is not clearly explained and even though there is a nice explanatory figure the text should be more clear.

A paragraph was added in order to make the feeding strategy more clear: 

Following the acclimation period, each experimental diet was randomly allocated to the fish tanks (two tanks/diet) and fed for 30 days (period 1) as described above. After 30 days, the feeding strategy were changed and new diets were fed for 7 days (period 2). As a control, one fish group received FM throughout the experiment. To assess whether C. utilis were able to counteract enteritis induced by SBM, four fish groups received SBM diets combined with different inclusions levels of CU (i.e. SBM25CU, SBM50CU, SBM100CU, SBM200CU) in period 1. One fish group received FM200CU in period 1 to evaluate if C. utilis in combination with FM alone would have an effect on the DI. In period 2, the ability of C. utilis to prevent SBMIE was assessed as the diet was changed to SBM in this group. Finally, three fish groups were fed SBM diet to induce SBMIE in period 1, and in period 2 the diets were changed to either FM, FM200CU and SBM200CU to evaluate if these diets were able to reverse SBMIE. The feeding strategy is illustrated in Fig. 1.

Subsequently L132-135 were changed to: 

At each sampling point (0, 7, 30 and 37 days), 8 fish per diet (4 fish per tank) were randomly selected and anaesthetized by immersion in 60 mg/l of tricaine methanesulfonate (MS-222, Sigma-Aldrich, MO, USA) and subsequently euthanized by a sharp blow to the head.

L200: The amount of tissue used for RNA isolation should be stated.

The sentence have been changed to: A small piece of DI tissue (approximately 0.5 cm) from FM, FM200CU, SBM and SBM200CU diet groups (8 fish/diet) at day 30 were subject to gene expression analysis.

L215: The concentration of the primers should be stated. Also, in the supplementary figure of the primer sequences the symbol for GDPH is GADPH, please unify the nomenclature.

The primer concentration has been added.

GDPH had been changed to GAPDH

L253: I think a reference to the salmon proteome should appear here.

The salmon proteome was added to the main text (https://www.uniprot.org/proteomes/?query=taxonomy:8030)

L269: Why a series of PCA? Were several analyses performed? If so, they should be described. Also, verify the use of analyses instead of analysis for the plural.

The description of the analysis was edited for a better understanding.

“An initial PCA was used as an unsupervised method to find the directions of maximum covariance among FM, SBM, SBM200CU and FM200CU was performed using the prcomp package in R to see the distribution of the proteins present in all the groups”.

L289: The average initial and final weight are mentioned (again I’m missing the errors), but there is no comparison among groups. Were there differences? Maybe a table should be included with the group values?

This was not a growth experiment. But to be sure that fish in all dietary treatments were eating, and growing, the feed intake was registered. Fish weights were only registered as bulk weight per tank at the start of the experiment, and as individual body weights for the fishes that were sampled in weeks 0, 1, 4 (period 1) and 5 (period 2). To avoid unnecessary disturbance, the fish left in the tank at the different time points were not taken out for weighing. Thus at the end of the experiment, there were very few fish left per tank. We feel that growth responses should therefore not be presented in a table. 

However, there were no significant differences due to dietary treatment when it comes to both feed intake nor growth. Thus we have added “nor growth rate” to make it more clear to the reader. The new sentence reads: “All groups of fish accepted their allocated diets and no significant differences were found in feed intake nor growth rate among dietary treatment”.

L303: “described in detail below” (remove the “ed”)

The authors don’t think it will be correct to remove “ed” in this context and want to leave it as it is. 

L293: section Histology: A panel of pictures showing the most significant changes (like the ones described in L308-312) would be nice. Also, how do the authors know exactly which cell types are present (L311-312) without specific markers? Some of these cells are difficult to identify in intestinal sections.

A panel of pictures (S2 Fig.) is included showing the overall differences between the groups but also variation within the SBM25 group.

Yes, it is indeed difficult to identify specific leucocytes in intestinal sections but not impossible. There was not made an effort to make a semi-quantitative assessment of the various cell types in the different diets, thus only the leucocytes that were in abundancy will be mentioned. Therefore, L309-312 will be reduced to: “There was an increased presence of connective tissue in the lamina propria and the increased infiltration of leucocytes consisted mainly of eosinophilic granule cells and to a lesser extent of lymphocytes.”

L323-324: Why where those groups excluded from the figure?

L323-324: these two SBM groups were similar with the SBM group presented in the figure and therefore omitted from the figure to generate equal groups. The individuals from both of these groups are now included making the SBM group n = 23.

L366-367: The difference mentioned here is not represented in the figure, in fact, regarding the figure statistics these groups were not different in this parameter.

The data for morphometric measurements and CD-densities were analyzed in JMP, but the figure was made in GraphPad. Running wrong statistical analysis in GraphPad resulted in wrong statistical differences expressed as “a”, “b” and “c” in the figure, and consequently, the wrong figure was submitted. The significant differences were presented in the result section and discussed in the text according to the statistics that were originally run in JMP. To simplify the text, the data set has been rerun in GraphPad and the correct figure has been submitted in the revised version. 

L240-247 have been updated to: 

“Non-parametric data from the histological evaluation were analyzed by Kruskal-Wallis followed by post hoc Dunn’s test with a comparison of mean rank. Shapiro-Wilk normality test was used to test the normal distribution of the data from morphometric analyses and T-cell density, and further were analyzed by one-way ANOVA followed by Tukey’s multiple comparisons test. Morphometric analyses and T-cell density analyses were performed at the individual level using the mean of measurements of between 2-6 simple folds per fish. Results of qPCR (means ± standard deviations) were analyzed using One-way ANOVA with Dunnett`s multiple comparison test (a < 0.0001). These analyses were performed in GraphPad Prism, version 7.00 and 8.0.1 (GraphPad Software Inc., San Diego, CA, USA).

L387: The sudden change in nomenclature does not simplify the reading of the data, it actually complicates it. The nomenclature should be maintained along the manuscript (including the figures). In addition, why were the groups with low yeast dose excluded from this analysis? This should be explained in detail, particularly because the work concludes that the low doses are the ones having more positive effects but there are no results about them in this part of the study.

Besides the explanation included in the main text, we included a extra legend on Fig 6 and 7, referring to the different groups and its nomenclature. The aim of the proteomic analysis was to identify potential biomarkers or protein express uniquely on the SBM induced enteritis groups. Therefore, we choose the extreme groups. Nevertheless, we did not observe unique proteins which can be used as markers. Then we focus on the proteins that were differently expressed, but present in all the groups. We agree with the Reviewers at it could be very interesting to run proteomic analysis considering the low dose, it will be considered in our future trials.

L403 and Fig6C: This heatmap seems to show differential expression values. If this is correlation I don’t understand against what each molecule is correlated. Please revise these results and explanations.

The heatmap was generated based on the pairwise comparisons across the four time points. As per your suggestion, to be simplified and easy for the audience to follow, we have re-analyzed and produced three heatmaps, with each diet treatment to the control (FM). The old heatmap figure is replaced with new heatmaps (Fig 6C).

L409: “Analyses were carried out first with….” Why first? Was there a second analysis?

The description of the analysis was edited as follow:

“PCA analysis was carried out first with all selected proteins across dietary groups, then PLS-DA multivariate analyses were performed to detect the proteins responsible for the differentiation between dietary groups (Fig 7D).”

L445: These inconsistencies should be further explained.

These inconsistencies have further been explained with:

Furthermore, Miadoková et al concluded in their study that the biological activity of glucomannan isolated form C. utilis dependent on the combined application with other biologically active compounds [56]. Also, the host itself can be the reason for this inconsistency. It has been demonstrated that the severity of SBMIE can differ between strains of rainbow trout [57], and that diet can induce variation in both the mucosa-associated and digesta-associated microbiota in Atlantic salmon [58, 59].

L473: Delete DI.

The discussion has been rewritten, thus this is not relevant in the new discussion. 

L473-475: The statements about the role of aqp8 in homoeostasis should be further supported.

L473-475 has been removed as our data does not support this statement. 

L489-492: I do not see the point of mentioning these particular roles of MHCI here. Maybe the authors are trying to make a point I do not understand.

This paragraph has been rewritten in the discussion. 

L494: One study or several? If one, change for “A previous study”. Otherwise use plural.

This line has been rewritten as part of the rearrangement of the discussion. 

L509: Downregulation in which tissue?

In the study performed by Grammes et al, a downregulation of LYME mRNA was observed in DI. Our study indicate that in DI inflammation there are high plasma levels of LYME. 

Figure 3: Scale bars should be added to the pictures.

Scale bars are added to the pictures

Figure 5: The fold change is relative to what?

The relative fold changes are calculated in relation to the FM group.

Reviewer #2: Review of PONE-D-19-15096 ” Candida utilis yeast as a functional protein source for Atlantic salmon (Salmo salar L.): Local intestinal tissue and plasma proteome responses”.

This manuscript describes the effects of using a yeast strain as a functional protein source to boost gut health in fish at risk of intestinal enteritis when fed soy bean-based meal. The study tackles a highly relevant problem and is generally very well written and clear. Overall, this study will provide an important contribution to the field and fit well into the PLoS one journal. Below I provide some general and more specific comments to help the authors improve the manuscript for publication.

General comments

I want to applaud the authors for performing such a holistic analysis opposed to using just one single method. This is very much needed and sets a great standard for future studies.

However, what I miss a little is an attempt to better link all the data sets together in a biological way. Could simply be through discussing results, but in a way to e.g. link the genes tested for expression responses with the proteins analyses all the way to the tissue histological data. Maybe add a conceptual figure with the entire path from DNA -> RNA -> proteins -> tissues and the highlight where you have generated data and results. As is, the different data types are treated quite independently and in a somewhat random order.

An attempt to link the data sets together in a biological way have been made in the rewritten discussion. 

For example, the discussion on lines 467-469 represents a good example of how gene expression data is integrated with insights from histology and immune response data.

Contrary, on lines 484-488 is an example where the protein plasma data is mainly discussed in isolation, whereas it would benefit from a stronger focus on the underling biological interactions with e.g. gene expression, histology and immune response data.

We agree with the Reviewers. Therefore the discussion have been rewritten.

Specific comments

L. 108: What was the fish vaccinated against? Was it a standard vaccination program or inly for a specific disease? Pease specify.

Farmed Atlantic salmon in Norway are routinely vaccinated as pre-smolts against a number of diseases. In this case, the Aquavac PD7 was used by Sørsmolt AS. 

We have added this information, thus the sentence now reads: “Vaccinated salmon (Aquavac PD7, MSD Animal Health, Bergen, Norway) were acquired from…”.

L. 158: Which and how many fish were used for this immunohistochemistry analysis? Was it the same 8 fish as above or new ones? Please specify.

The same 8 fish from each diet group were used for histology, immunohistochemistry and qPCR. L158-159 has been changed in order to make this clearer: 

Histological sections of DI from the fish sampled at day 30 (8 fish/diet), prepared as described above, was subjected for immunohistochemical analysis, and the following diet groups were included: FM, FM200CU, SBM, SBM25CU and SBM200CU.

L. 224: Do you mean “… as index” instead of “… an index”?

L224 changed to “… an index”

L. 229 + 231 + throughout: Be consistent whether or not you add a space between the degree symbol and “C” when listing temperature values.

When listing temperature values, all have been changed to °C with no space between degree symbol and “C”. 

L. 253: Please provide a reference for the salmon proteome reference database.

The salmon proteome as been included in material and method

L. 262-263: Can you clarify this description as it sounds like all individuals are already part of "a cluster" when you begin. So, when does the model know to end the analysis? Is it e.g. trying to minimise the number of clusters based on some preset value? This remains unclear as currently written.

We agree and the sentence have been modifies as follow:

“The clustering results are presented in the form of a heatmap, with levels of protein expression across the three dietary groups (SBM (D2), SBM200CU (D6) and FM200CU (D7) compared to the control FM (D1). Hierarchical clustering was performed with the hclust function in R package stat. UniprotKB database was used for functional annotation of the proteins.”

L. 275: Add space before “[31]”.

Space has been added before “[31]”

L. 388-389: It is not clear to me how this "simplifies" except using fewer letters. The figures have plenty of space to write out the FM and SBM names respectively, and these are much more informative than having to cross read between the text here and the figure to know what D1-D4 represents. I would prefer to just keep the FM and SBM based names in the figures as well. Same for Fig. 7.

Besides the explanation included in the main text, we included an extra table on Fig 6 and 7, referring to the different groups and its nomenclature. 

L. 438-444: Could this maybe also be due to individual responses among fish, and that these responses are partly controlled by the different genotypes among individuals? For example, if host genotype selects for different gut-microbiota communities, one could speculate that such different gut environments will differ in their ability to respond to the provided C. utilis strains? I agree this is very speculative, but the explanation do warrant mention as opposed to only focusing on how external parameters could explain the observed variation in gut responses; which seems to be the standard way of interpreting these types of gut data.

This is a good point which have already been briefly mentioned/discussed in the following sentences in the manuscript: 

“Also, the host itself can be the reason for this inconsistency. It has been demonstrated that the severity of SBMIE can differ between strains of rainbow trout (Venold 2012), and that diet can induce variation in both the mucosa-associated and digesta-associated microbiota in Atlantic salmon (Gajardo 2016, 2017)”.

However, as we do not have the actual genotypes of these salmon (they were purchased from Sørsmolt AS), a further discussion of this point would be just speculation.

L. 503: re-write to "proteins with significant over expression in ... ". The proteins themselves are not significant, it is their expression level that is.

This line has been rewritten as part of the rearrangement of the discussion. 

Figure 2: Please add labels for all Y axes.

The label “Scale” is added for all Y axes.

Figure 3: Please clarify the different colors in the figure images. Are CD8 positive cells the purple colored ones? Or are both CD3 and CD8 colored brown? If so, what are the purple stains?

Color development of both CD3 and CD8 was performed by 3,3’-diaminobenzidine, therefore both CD3 and CD8 positive cells are labeled with brown color. As the detection of these cells were run in two different assays where specific antibody was applied, we can count positive CD3-labelled cells when antibody towards CD3 was applied and we can count positive CD8-labeled cells when antibody towards CD8 was applied. In both cases, the slide was counterstained with hematoxylin that gives a purple color in order to visualize to the structure of the rest of the intestinal mucosa. 

Figure 4: Discuss whether the low sample size (n=7) may cause type-2 errors for the non-significant comparisons?

The sample size was in fact n=8, except for the SBM group, but we agree that this is indeed a low sample size. We would of course have preferred to have a higher number of fish, but the experimental set-up, number of available tanks and limit of biomass in each tank did not allow for a higher number. A larger sample size, e.g., by one more tank per dietary treatment (i.e., 3 tanks instead of 2) could have strengthened our results. However, since we have measured/analyzed many different parameters in the same individual fish, and these analyses do support each other, we still feel confident with these results.

Figure 5: Add both a and b letters between the two diet categories in the plots where there are significant differences.

Letter a and b were added between the two diets where there are a significant differences 

Also, same concern as for figure 4 about regarding lack of statistical power with the relatively low sample sizes. Was any prior power analyses performed to inform the used sample sizes? There are some differences in e.g. plot C that could potentially be significant with more statistical power, which could for example be obtained through an increased sample size.

Unfortunately, the wrong figure was submitted. The correct figure is submitted in the revised version, and here, the differences are significant.

Reviewer #3: General Impressions:

In this manuscript, the authors performed a feeding trial on Atlantic salmon in which fish were fed fish meal or soybean meal-based diets that were supplemented with varying amounts of Canadis utilis yeast. Changes in intestinal morphology and the plasma proteome of the fish were monitored at several points in the study in order to monitor both intestinal and systemic effects of the different diets.

Overall, the experimental design appears valid and the data collection and analysis of the histological data is appropriate. However, in the eyes of this reviewer, the analysis of the plasma proteomics data could be improved to allow for more meaningful interpretation of the data. The current methodology does not provide significant evidence that the observed changes in the plasma proteome are related to the differences in diet. However, after the authors address the following comments, the manuscript would likely be suitable for publication.

Major Comments:

The authors relied primarily on Spearman’s correlation coefficients to identify proteins of interest from the plasma proteomics data. Therefore, the authors are essentially looking for cases where protein abundances do or do not correlate across the different diets. However, since there is a clear control in the experiment (fish meal-based diet), it seems that comparing all protein abundance to that control would be more informative. Then, emphasis would be placed on proteins that have ‘abnormal’ expression relative to the fishmeal fed fish. This would also allow for volcano plots to be generated and included in the manuscript, which would aid reader understanding.

Yes, we re-analyzed the data. According to your suggestions, we have generarted the volcano plots and heat maps, by comparing each diet treatment to the control (FM). The old heatmap figure is replaced with new heatmaps (Fig 6C) and new volcano plots were added (Fig 7A, 7B, 7C).

• Further description of how the quantitative values used in the proteomics analysis is needed. It appears that precursor abundance (from the MS1 scans) were used, but that is not explicitly stated. If that is the case, was peak intensity or peak area used? Also, was the ‘match between runs’ feature enabled in MaxQuant?

We performed the anlysis in Mascot and MaxQuant, however, we used the LFQ on MS2 level (Spectral count) and not on MS1 (Peak areas) as in MaxQuant to perform the analysis. The MS raw file were analyzed in both Mascot and Scaffold to get higher confidence in the protein identified. 

• A table showing all of the proteins identified in the plasma analyses and their quantitative values should be provided in the supplement. Ideally, fold change values should also be included. It’s possible that such a table was uploaded to PRIDE, but the PXD number provided does not bring up any results when searched. This is likely because the dataset has not been publicly released but the login info should be provided to reviewers.

Yes, an excel sheet containing all the proteins identified with their quantitative values and fold change values are submitted as a supplementary table (S3 Table).

• It is concerning that the fishmeal-based and soybean-based diets don’t cluster in either the heatmap or the PCA plots. It seems likely that such large differences in the feed would causes systemic changes in the blood which would lead to high correlation between the D1 and D7 samples, but that was not observed. This suggests that either the analysis method was flawed, or that individual variation was too large in order to adequately detect differences from the feed.

It is important to mention that we did not observe severe enteritis on the SBM fed fish, which might have influenced the results on plasma proteome. On the other hand, as mentioned by Reviewer 3, we observed a very big variation between individual. This is a persistent concern in feeding trial with fish, since there is not an available method to measure individual feed intake as in mammals. Therefore we use each tank as unit and calculate feed intake per tank. Although we perform histology to verify the degree of enteritis of each fish used in the analysis, we are not sure how much of the antinutrient (SBM) each fish ingested. Our main aim goal was to identify proteins uniquely expressed in each group, but it seems at SBM enteritis is restricted to a local inflammation (distal intestine) instead of a systemic inflammation.

• Although this reviewer is not well versed in the VIP method used here, it seems that this technique is likely to put too much emphasis on proteins that are correlated purely by chance. Further justification of this technique should be included, or the analyses should be removed.

In the initial PCA plot (Fig 7D), there is significant overlap of dietary groups, which leads to difficulty in discerning which proteins are responsible for the changes seen between the dietary groups. To address this problem, we used a known approach (Eriksson L, Umetrics AB (2006) Multi- and Megavariate Data Analysis, Part 1, Basic Principles and Applications: Umetrics AB.) using Partial Least Squares – Discriminant Analysis (PLS-DA) Variable Importance Projection (VIP) scores to highlight the significant changes between these groups. These VIP scores are normally used for variable selection as heat maps combined with group difference proteins to highlight the significant differences in dietary groups. We don’t think, this VIP proteins are correlated purely by chance, as in the discussion, we have highlighted about the role of these proteins on immune system and how they are related with SBM enteritis.

• It would be helpful if the reviewers included a few representative images of the changes in morphology observed between the groups that are summarized in Fig. 2. This could be included as an additional supplemental figure.

We added Fig. 2S, where morphologic changes are presented due to the diets.

Minor Comments:

• L252: The fragment ion tolerance used here is much too large for orbitrap data. A tolerance of 0.02 Da would be more appropriate.

We appreciate the comment of Reviewer 3, the MS analysis was performed at Proteomic Core Facility of University of Oslo. But the recommendation will be taken into account in our future analysis.

• L253: Where was the salmon proteome database derived from? Was it a public repository such as uniprot? If so, when was it downloaded?

The reference salmon proteome was download from a public repository UniProt, as mention in the main text. The date of downloaded was December 2017.

• L316: No data is shown in this figure where SBM + C. utilis was fed first, followed by FM. This sentence says otherwise.

All the groups are included in the new figure.

• Fig. 4 caption – L371: What is meant hear by ‘data are expressed as mean � SD for each individual’? Isn’t each point just representative of the mean for each fish? This should be clarified.

Data are expressed as the mean for all individuals � SD.

• L407-408: It doesn’t make sense to say that proteins ‘belong’ to a specific domain. Please revise.

Yes, we revised the sentence and changed to “most of the proteins have signal protein domain”.

• Fig. 7 caption – L424: What do the colors in panel C represent? Also, why are proteins with VIP scores < 2 included here?

The colors in the panel C represents the protein abundance at 4 dietary treatments. Red represents higher and dark green represents lower abundance. This sentence is added to the figure legend. We have replaced the figure with new one showing the proteins only with VIP >2. 

• L473: Remove ‘DI.’

This line has been rewritten as part of the rearrangement of the discussion. 

• L476: Change considering to considered.

This line has been rewritten as part of the rearrangement of the discussion.

---

## [Decision Letter · Decision Letter 1]

13 Sep 2019

PONE-D-19-15096R1

Candida utilis yeast as a functional protein source for Atlantic salmon (Salmo salar L.): Local intestinal tissue and plasma proteome responses

PLOS ONE

Dear Dr. Øverland,

Thank you for submitting your manuscript to PLOS ONE. After careful consideration, we feel that it has merit but does not fully meet PLOS ONE’s publication criteria as it currently stands. Therefore, we invite you to submit a revised version of the manuscript that addresses the points raised during the review process.

While all three reviewers consider the general response to comments satisfactory and that the manuscript is significantly improved, Reviewer 3 has underlined new concerns regarding the proteomic data analysis. Based on the possibility that the conclusion reached in the manuscript may be based on invalid interpretation of the results. The reviewer kindly suggests many viable options to overcome this conundrum, and I am therefore confident that the matter can be resolved.

Specific comments by both reviewers 2 and 3 should also be addressed prior to publication.

We would appreciate receiving your revised manuscript by Oct 28 2019 11:59PM. To enhance the reproducibility of your results, we recommend that if applicable you deposit your laboratory protocols in protocols.io, where a protocol can be assigned its own identifier (DOI) such that it can be cited independently in the future. For instructions see: http://journals.plos.org/plosone/s/submission-guidelines#loc-laboratory-protocols

We look forward to receiving your revised manuscript.

Kind regards,

Annie Angers, Ph.D.

Academic Editor

PLOS ONE

Reviewers' comments:

Reviewer's Responses to Questions

**Comments to the Author**

1. If the authors have adequately addressed your comments raised in a previous round of review and you feel that this manuscript is now acceptable for publication, you may indicate that here to bypass the “Comments to the Author” section, enter your conflict of interest statement in the “Confidential to Editor” section, and submit your "Accept" recommendation.

Reviewer #1: All comments have been addressed

Reviewer #2: (No Response)

Reviewer #3: (No Response)

2. Is the manuscript technically sound, and do the data support the conclusions?

Reviewer #1: Yes

Reviewer #2: Yes

Reviewer #3: Partly

3. Has the statistical analysis been performed appropriately and rigorously? 

Reviewer #1: Yes

Reviewer #2: Yes

Reviewer #3: Yes

4. Have the authors made all data underlying the findings in their manuscript fully available?

Reviewer #1: Yes

Reviewer #2: Yes

Reviewer #3: Yes

5. Is the manuscript presented in an intelligible fashion and written in standard English?

Reviewer #1: Yes

Reviewer #2: Yes

Reviewer #3: Yes

6. Review Comments to the Author

Reviewer #1: (No Response)

Reviewer #2: I have now been through this revised version of the manuscript. I am generally happy with all the authors responses to my original review. So I am happy to hereby recommend the manuscript for publication.

But I do have a few minor comments to help with some final improvements:

1)

Regarding the Y-axes labels on figure 2, it may be more appropriate to label them "Change" rather than "Scale" that seems not very informative.

2)

I suggest that the authors either delete or re-write the very kast sentence of the discussion:

"Further research is needed to evaluate the impact of yeast strain and the fermentation

and/or down-stream processing conditions of the yeast on functional properties in relation to gastro

intestinal health and systemic responses."

As is, this is not very informative. Consider adding some more concrete suggestions as to how such "Further research" can be designed to better help move this entire research field and add more values to readers of the paper.

Reviewer #3: The authors addressed a number of concerns raised by the reviewers and the manuscript has been significantly improved. Unfortunately, this reviewer still has significant concerns about the analysis of the proteomics data and the conclusions that are drawn from this data.

In the original manuscript, it was not made clear that the authors were utilizing spectral counting to obtain quantitative information from the proteomics data. This should have been stated in the manuscript, and needs to be added. The spectral counting method has been phased out in the field due to its low reproducibility and accuracy. When using a high resolution, accurate mass instrument such as the Q Exactive that was used here, MS1 intensity-based quantitation provides far superior quantitative values than does spectral counting. The reasons for this are well described in the literature, but are primarily the result of the somewhat stochastic nature of the data collection when data-dependent acquisition strategies are employed (as was done in this study). The fact that an MS2 mass tolerance was used for searching that was more than an order of magnitude higher than it should have been (0.8 Da rather than 0.02 Da) further reduces the quantitative accuracy of the spectral counting strategy used here.

For the reasons described above, the quantitative values presented should be treated with caution. The proteins shown to be differentially expressed in the volcano plots (Fig. 7) are likely valid, as this analysis incorporates both the calculated fold change, as well as a p-value, which incorporates other variables such as inter-sample variability. Unfortunately, many of the proteins that the authors focus on in the discussion (such as HMP, LYME, and HIIN) were not shown to be differentially expressed by this analysis, they simply didn't correlate well to the control. This alone is not adequate evidence of a change in protein abundance.

With these concerns in mind, this reviewer suggests the authors choose one of 3 options:

1. Reanalyze the data using MS1 intensity-based quantitation (as can be completed in MaxQuant or other software packages such as Scaffold Q+, Proteome Discoverer, etc.).

2. Focus the discussion only on proteins that showed differential abundance in the volcano plot analyses (Fig. 7)

3. Remove the proteomics data entirely and focusing on the gene expression, morphological, and other data presented in the manuscript.

Any of the above approaches would be valid and would make the publication suitable for publication in the opinion of this reviewer.

Specific Comments:

L262: The date the database was downloaded should also be included.

L403: Don't say 'expressed' here. The proteins were likely expressed in the samples, they just couldn't be detected by the mass spectrometer.

Figure 6: The Spearman's correlations done here do not provide sufficient evidence to say a protein is changing abundance. Panel C should be removed from the manuscript as well as all discussion of the results presented in this panel.

7. PLOS authors have the option to publish the peer review history of their article (what does this mean?). If published, this will include your full peer review and any attached files.

Reviewer #1: No

Reviewer #2: No

Reviewer #3: No

---

## [Author Response · Author response to Decision Letter 1]

28 Oct 2019

6. Review Comments to the Author

Reviewer #1: (No Response)

Reviewer #2: I have now been through this revised version of the manuscript. I am generally happy with all the authors responses to my original review. So I am happy to hereby recommend the manuscript for publication.

But I do have a few minor comments to help with some final improvements:

1)

Regarding the Y-axes labels on figure 2, it may be more appropriate to label them "Change" rather than "Scale" that seems not very informative.

The label on the Y-axis has been changed from "scale" to "change" according to the reviewer’s request.

2)

I suggest that the authors either delete or re-write the very kast sentence of the discussion:

"Further research is needed to evaluate the impact of yeast strain and the fermentation and/or down-stream processing conditions of the yeast on functional properties in relation to gastro intestinal health and systemic responses."

As is, this is not very informative. Consider adding some more concrete suggestions as to how such "Further research" can be designed to better help move this entire research field and add more values to readers of the paper.

The last sentence has been deleted as the discussion has been changed. 

Reviewer #3: The authors addressed a number of concerns raised by the reviewers and the manuscript has been significantly improved. Unfortunately, this reviewer still has significant concerns about the analysis of the proteomics data and the conclusions that are drawn from this data.

In the original manuscript, it was not made clear that the authors were utilizing spectral counting to obtain quantitative information from the proteomics data. This should have been stated in the manuscript, and needs to be added. The spectral counting method has been phased out in the field due to its low reproducibility and accuracy. When using a high resolution, accurate mass instrument such as the Q Exactive that was used here, MS1 intensity-based quantitation provides far superior quantitative values than does spectral counting. The reasons for this are well described in the literature, but are primarily the result of the somewhat stochastic nature of the data collection when data-dependent acquisition strategies are employed (as was done in this study). The fact that an MS2 mass tolerance was used for searching that was more than an order of magnitude higher than it should have been (0.8 Da rather than 0.02 Da) further reduces the quantitative accuracy of the spectral counting strategy used here.

For the reasons described above, the quantitative values presented should be treated with caution. The proteins shown to be differentially expressed in the volcano plots (Fig. 7) are likely valid, as this analysis incorporates both the calculated fold change, as well as a p-value, which incorporates other variables such as inter-sample variability. Unfortunately, many of the proteins that the authors focus on in the discussion (such as HMP, LYME, and HIIN) were not shown to be differentially expressed by this analysis, they simply didn't correlate well to the control. This alone is not adequate evidence of a change in protein abundance.

With these concerns in mind, this reviewer suggests the authors choose one of 3 options:

1. Reanalyze the data using MS1 intensity-based quantitation (as can be completed in MaxQuant or other software packages such as Scaffold Q+, Proteome Discoverer, etc.).

2. Focus the discussion only on proteins that showed differential abundance in the volcano plot analyses (Fig. 7) 

3. Remove the proteomics data entirely and focusing on the gene expression, morphological, and other data presented in the manuscript.

Any of the above approaches would be valid and would make the publication suitable for publication in the opinion of this reviewer.

We are grateful for the specificity and professional comments of reviewer 3. We decided to take option number 1, which means at we reanalyzed the entire dataset using MaxQuant and Perseus. The LFQ intensities were exported to an excel sheet. The log-transformed LFQ values were analyzed using the pipeline described in material and methods. We created volcano plots, comparing the protein pattern of each dietary treatment with the control diet. This comparison resulted in 9, 12 and 10 significantly expressed protein when the FM was compared with SBM, SBM200CU and FM200CU, respectively. Although option 1 was more complicated, we are very grateful for the comments, which helped us to increase the reliability of our results.

Specific Comments:

L262: The date the database was downloaded should also be included.

We used the latest salmon proteome available and the date is included in the manuscript

L403: Don't say 'expressed' here. The proteins were likely expressed in the samples, they just couldn't be detected by the mass spectrometer.

The sentence was changed as marked in the manuscript.

Figure 6: The Spearman's correlations done here do not provide sufficient evidence to say a protein is changing abundance. Panel C should be removed from the manuscript as well as all discussion of the results presented in this panel.

In accordance with the reviewer’s advice, Panel C in Figure 6 has been removed from the manuscript and the discussion revised accordingly.

---

## [Decision Letter · Decision Letter 2]

11 Nov 2019

PONE-D-19-15096R2

Candida utilis yeast as a functional protein source for Atlantic salmon (Salmo salar L.): Local intestinal tissue and plasma proteome responses

PLOS ONE

Dear Dr. Øverland,

Thank you for submitting your manuscript to PLOS ONE. After careful consideration, we feel that it has merit but does not fully meet PLOS ONE’s publication criteria as it currently stands. Therefore, we invite you to submit a revised version of the manuscript that addresses the points raised during the review process.

The manuscript is much improved and almost ready for publication. However, the final concerns raised by Reviewer 3 still need to be addressed, and minor adjustments to the text are also needed.

We would appreciate receiving your revised manuscript by Dec 26 2019 11:59PM. To enhance the reproducibility of your results, we recommend that if applicable you deposit your laboratory protocols in protocols.io, where a protocol can be assigned its own identifier (DOI) such that it can be cited independently in the future. For instructions see: http://journals.plos.org/plosone/s/submission-guidelines#loc-laboratory-protocols

We look forward to receiving your revised manuscript.

Kind regards,

Annie Angers, Ph.D.

Academic Editor

PLOS ONE

Additional Editor Comments (if provided):

- Please address the possible data discrepancies between table S3 and table 2 referred to by Reviewer 3;

- Make sure that proteins that were not found in a replicate are not given a value of zero but treated as missing in your analysis (or clearly state that you did not use 0 in your analysis)

- Make the minor adjustments to text suggested by Reviewer 3.

Reviewers' comments:

Reviewer's Responses to Questions

**Comments to the Author**

1. If the authors have adequately addressed your comments raised in a previous round of review and you feel that this manuscript is now acceptable for publication, you may indicate that here to bypass the “Comments to the Author” section, enter your conflict of interest statement in the “Confidential to Editor” section, and submit your "Accept" recommendation.

Reviewer #3: (No Response)

2. Is the manuscript technically sound, and do the data support the conclusions?

Reviewer #3: Yes

3. Has the statistical analysis been performed appropriately and rigorously? 

Reviewer #3: Yes

4. Have the authors made all data underlying the findings in their manuscript fully available?

Reviewer #3: Yes

5. Is the manuscript presented in an intelligible fashion and written in standard English?

Reviewer #3: Yes

6. Review Comments to the Author

Reviewer #3: General Impressions:

The authors have clearly addressed the concerns raised in the previous review, and the reanalysis of the proteomics data has dramatically improved the quality of the manuscript. I greatly appreciate the authors hard work and dedication to providing the highest quality data possible. I have noticed one discrepancy in the data analysis that should be addressed in the final version of the manuscript (described below), but I feel that the manuscript is otherwise ready for publication. I’ve also included a few very minor comments that I feel will improve the readability of the publication.

Major Comments:

The authors state that only protein that were identified in at least two of the four replicates were used for quantitative analysis. However, when looking at Table S3 and comparing that to the results presented in Table 2, there seems to be some discrepancies. For example, C0HAL2 (elongation factor 1-alpha) is reported as having a very significant fold-change between the FM (D1) and SBM (D2) in Table 2, yet no quantitative values for this protein are reported in the D2 group in Table S3. Why is this? Please ensure that your filtering is working as intended.

Additionally, proteins that were not identified in a replicate should not be assigned a quantitative value of 0. Instead, these values should be treated as missing (null), or missing value imputation should be used to provide baseline quantitative values. It’s difficult to determine how such values were treated in the analysis, and that should be stated outright.

As an aside for future experiments, use of the ‘match between runs’ feature in MaxQuant should dramatically decrease the number of missing values present in the dataset.

Minor Comments:

Line 241: It seems as if part of this sentence was accidentally removed as there is no mention of the addition of trypsin.

Line 243: Another sentence or two very briefly describing the column and MS parameters that were used would be useful here.

Line 261: Was a fragment mass tolerance of 0.8 used in the reprocessed data as well or was this a carryover from the previous version of the manuscript? As mentioned previously, a much tighter tolerance (~0.02 Da) would be more appropriate.

Lines 276-277: Saying that the average weights increased does not mean that every fish gained weight. Consider rephrasing.

Table 2: It would be nice if you included the p-value for each protein here as well.

Figure 6: As a note, the ‘presence’ of a protein in one sample and not in another is much more likely a result of the stochastic nature of the MS data collection rather than a protein actually being present in the sample or not. Therefore, comparisons such as the one shown here are often not very informative.

7. PLOS authors have the option to publish the peer review history of their article (what does this mean?). If published, this will include your full peer review and any attached files.

Reviewer #3: No

---

## [Author Response · Author response to Decision Letter 2]

28 Nov 2019

6. Review Comments to the Author

Reviewer #3: General Impressions:

The authors have clearly addressed the concerns raised in the previous review, and the reanalysis of the proteomics data has dramatically improved the quality of the manuscript. I greatly appreciate the authors hard work and dedication to providing the highest quality data possible. I have noticed one discrepancy in the data analysis that should be addressed in the final version of the manuscript (described below), but I feel that the manuscript is otherwise ready for publication. I’ve also included a few very minor comments that I feel will improve the readability of the publication.

Major Comments:

The authors state that only protein that were identified in at least two of the four replicates were used for quantitative analysis. However, when looking at Table S3 and comparing that to the results presented in Table 2, there seems to be some discrepancies. For example, C0HAL2 (elongation factor 1-alpha) is reported as having a very significant fold-change between the FM (D1) and SBM (D2) in Table 2, yet no quantitative values for this protein are reported in the D2 group in Table S3. Why is this? Please ensure that your filtering is working as intended.

 We follow the criteria “at least two peptides in at least half of the replicates” per group. It means to asseverate that a protein is present in one or all the groups, should be at least in two samples. As an example, protein C0HAL2 is detected in all replicated of diet 1, in one of the replicates on diet 6 and in three out of the four replicates on diet 7. It was not detected on diet 2, therefore the significant difference between diet 1 and diet 2. From the total 158 proteins, 126 proteins were present in all dietary treatment, however the other 32 proteins were present in either one, two or three of the treatment, not in all, as stated in the manuscript. 

Additionally, proteins that were not identified in a replicate should not be assigned a quantitative value of 0. Instead, these values should be treated as missing (null), or missing value imputation should be used to provide baseline quantitative values. It’s difficult to determine how such values were treated in the analysis, and that should be stated outright.

 The function missing value imputation was used for the quantitative analysis. The imputed value is shown in Table S3 and it is mentioned in the manuscript as follow:

“Protein raw data were transferred to log normalization; missing value imputation was used to replace the not identified proteins on the quantitative analysis and then performed on autoscaled data (mean-centered and divided by the standard deviation of each variable) [29].”

As an aside for future experiments, use of the ‘match between runs’ feature in MaxQuant should dramatically decrease the number of missing values present in the dataset.

 We are very grateful for the suggestions, we have been learning a lot in this process so we are confident that in future experiments we will take into practice the knowledge acquired.

Minor Comments:

Line 241: It seems as if part of this sentence was accidentally removed as there is no mention of the addition of trypsin.

 Yes, it was an editing mistake. The sentence has been rewritten as follow:

“Subsequently, the proteins were digested with 10 μg trypsin (Promega, sequencing grade) overnight at 37°C. The digestion was stopped by adding 5 μL 50% formic acid and the generated peptides were purified using a ZipTip C18 (Millipore, Billerica, MA, USA) according to the manufacturer’s instructions, and dried using a Speed Vac concentrator (Concentrator Plus, Eppendorf, Hamburg, Germany)”

Line 243: Another sentence or two very briefly describing the column and MS parameters that were used would be useful here.

 The information required was added into the text

“The tryptic peptides were dissolved in 10 µL 0.1% formic acid/2% acetonitrile and 5 µL analyzed using an Ultimate 3000 RSLCnano-UHPLC system connected to a Q Exactive mass spectrometer (Thermo Fisher Scientiﬁc, Bremen, Germany) equipped with a nanoelectrospray ion source. For liquid chromatography separation, an Acclaim PepMap 100 column (C18, 2 µm beads, 100 Å, 75 μm inner diameter, 50 cm length) (Dionex, Sunnyvale CA, USA) was used. The mass spectrometer was operated in the data-dependent mode to automatically switch between MS and MS/MS acquisition. Survey full scan MS spectra (from m/z 400 to 2,000) were acquired with the resolution R = 70,000 at m/z 200, after accumulation to a target of 1e5. The maximum allowed ion accumulation times were 60 ms”

Line 261: Was a fragment mass tolerance of 0.8 used in the reprocessed data as well or was this a carryover from the previous version of the manuscript? As mentioned previously, a much tighter tolerance (~0.02 Da) would be more appropriate.

 This corresponded to the previous analysis, the new analysis was performed using:

“The fragment mass tolerance for the MS1 was 6 ppm and the fragment mass tolerance for the MS2 was 20 ppm”. 

This information was added into the text

Lines 276-277: Saying that the average weights increased does not mean that every fish gained weight. Consider rephrasing.

 The sentence was rephrased as follow:

 “The average initial weight was 526 g and the average final 277 weight was 667 g on day 37, considering that the weight was measured as bulk, this indicates that in general fish gained weight during the experimental period”

Table 2: It would be nice if you included the p-value for each protein here as well.

 We have included the p-value in Table 2 as required by Reviewer 3.

Figure 6: As a note, the ‘presence’ of a protein in one sample and not in another is much more likely a result of the stochastic nature of the MS data collection rather than a protein actually being present in the sample or not. Therefore, comparisons such as the one shown here are often not very informative.

 We are grateful for the comments and it will be considered in our future experiments.

---

## [Editor Report · Decision Letter 3]

9 Dec 2019

Candida utilis yeast as a functional protein source for Atlantic salmon (Salmo salar L.): Local intestinal tissue and plasma proteome responses

PONE-D-19-15096R3

Dear Dr. Øverland,

We are pleased to inform you that your manuscript has been judged scientifically suitable for publication and will be formally accepted for publication once it complies with all outstanding technical requirements.

With kind regards,

Annie Angers, Ph.D.

Academic Editor

PLOS ONE

Additional Editor Comments (optional):

Thank you for the extensive revisions to the manuscript. I am sorry that the processed took so long, but I feel that the improved version was worth the efforts.
---

## [Editor Report · Acceptance letter]

12 Dec 2019

PONE-D-19-15096R3 

*Candida utilis* yeast as a functional protein source for Atlantic salmon (*Salmo salar* L.): Local intestinal tissue and plasma proteome responses 

Dear Dr. Øverland:

I am pleased to inform you that your manuscript has been deemed suitable for publication in PLOS ONE. Congratulations! Your manuscript is now with our production department. 

With kind regards,

on behalf of

Dr. Annie Angers 

Academic Editor

PLOS ONE